# CONVERGENCE OF GRADIENT METHODS ON BILINEAR ZERO-SUM GAMES

**Guojun Zhang & Yaoliang Yu**
Department of Computer Science
University of Waterloo
Vector Institute
{guojun.zhang,yaoliang.yu}@uwaterloo.ca

## ABSTRACT

Min-max formulations have attracted great attention in the ML community due to the rise of deep generative models and adversarial methods, while understanding the dynamics of gradient algorithms for solving such formulations has remained a grand challenge. As a first step, we restrict to bilinear zero-sum games and give a systematic analysis of popular gradient updates, for both simultaneous and alternating versions. We provide exact conditions for their convergence and find the optimal parameter setup and convergence rates. In particular, our results offer formal evidence that alternating updates converge "better" than simultaneous ones.

## 1 INTRODUCTION

Min-max optimization has received significant attention recently due to the popularity of generative adversarial networks (GANs) (Goodfellow et al., 2014), adversarial training (Madry et al., 2018) and reinforcement learning (Du et al., 2017; Dai et al., 2018), just to name some examples. Formally, given a bivariate function $f(\boldsymbol{x}, \boldsymbol{y})$, we aim to find a *saddle point* $(\boldsymbol{x}^*, \boldsymbol{y}^*)$ such that

$$f(\boldsymbol{x}^*, \boldsymbol{y}) \leq f(\boldsymbol{x}^*, \boldsymbol{y}^*) \leq f(\boldsymbol{x}, \boldsymbol{y}^*), \ \forall \boldsymbol{x} \in \mathbb{R}^n, \ \forall \boldsymbol{y} \in \mathbb{R}^n. \tag{1.1}$$

Since the beginning of game theory, various algorithms have been proposed for finding saddle points (Arrow et al., 1958; Dem'yanov & Pevnyi, 1972; Gol'shtein, 1972; Korpelevich, 1976; Rockafellar, 1976; Bruck, 1977; Lions, 1978; Nemirovski & Yudin, 1983; Freund & Schapire, 1999). Due to its recent resurgence in ML, new algorithms specifically designed for training GANs were proposed (Daskalakis et al., 2018; Kingma & Ba, 2015; Gidel et al., 2019b; Mescheder et al., 2017). However, due to the inherent non-convexity in deep learning formulations, our current understanding of the convergence behaviour of new and classic gradient algorithms is still quite limited, and existing analysis mostly focused on bilinear games or strongly-convex-strongly-concave games (Tseng, 1995; Daskalakis et al., 2018; Gidel et al., 2019b; Liang & Stokes, 2019; Mokhtari et al., 2019b). Non-zero-sum bilinear games, on the other hand, are known to be PPAD-complete (Chen et al., 2009) (for finding approximate Nash equilibria, see e.g. Deligkas et al. (2017)).

In this work, we study bilinear zero-sum games as a first step towards understanding general min-max optimization, although our results apply to some simple GAN settings (Gidel et al., 2019a). It is well-known that certain gradient algorithms converge linearly on bilinear zero-sum games (Liang & Stokes, 2019; Mokhtari et al., 2019b; Rockafellar, 1976; Korpelevich, 1976). These iterative algorithms usually come with two versions: *Jacobi* style updates or *Gauss–Seidel* (GS) style. In a Jacobi style, we update the two sets of parameters (i.e., $\boldsymbol{x}$ and $\boldsymbol{y}$) *simultaneously* whereas in a GS style we update them *alternatingly* (i.e., one after the other). Thus, Jacobi style updates are naturally amenable to parallelization while GS style updates have to be sequential, although the latter is usually found to converge faster (and more stable). In numerical linear algebra, the celebrated Stein–Rosenberg theorem (Stein & Rosenberg, 1948) formally proves that in solving certain linear systems, GS updates converge *strictly* faster than their Jacobi counterparts, and often with a larger set of convergent instances. However, this result does not readily apply to bilinear zero-sum games.

Our main goal here is to answer the following questions about solving bilinear zero-sum games:

- When exactly does a gradient-type algorithm converge?

Table 1: Comparisons between Jacobi and Gauss–Seidel updates. The second and third columns show when exactly an algorithm converges, with Jacobi or GS updates. The last column shows whether the convergence region of Jacobi updates is contained in the GS convergence region.

| Algorithm | Jacobi | Gauss–Seidel | Contained? |
|---|---|---|---|
| GD | diverges | limit cycle | N/A |
| EG | Theorem 3.2 | Theorem 3.2 | if $\beta_1 + \beta_2 + \alpha^2 < 2/\sigma_1^2$ |
| OGD | Theorem 3.3 | Theorem 3.3 | yes |
| momentum | does not converge | Theorem 3.4 | yes |

Table 2: Optimal convergence rates. In the second column, $\beta_*$ denotes a specific parameter that depends on $\sigma_1$ and $\sigma_n$ (see equation 4.2). In the third column, the linear rates are for large $\kappa$. The optimal parameters for both Jacobi and Gauss–Seidel EG algorithms are the same. $\alpha$ denotes the step size ($\alpha_1 = \alpha_2 = \alpha$), and $\beta_1$ and $\beta_2$ are hyper-parameters for EG and OGD, as given in §2.

| Algorithm | $\alpha$ | $\beta_1$ | $\beta_2$ | Rate exponent | Comment |
|---|---|---|---|---|---|
| EG | $\sim 0$ | $2/(\sigma_1^2 + \sigma_n^2)$ | $\beta_1$ | $\sim 1 - 2/\kappa^2$ | Jacobi and Gauss–Seidel |
| Jacobi OGD | $2\beta_1$ | $\beta_*$ | $\beta_1$ | $\sim 1 - 1/(6\kappa^2)$ | $\beta_1 = \beta_2 = \alpha/2$ |
| GS OGD | $\sqrt{2}/\sigma_1$ | $\sqrt{2}\sigma_1/(\sigma_1^2 + \sigma_n^2)$ | $0$ | $\sim 1 - 1/\kappa^2$ | $\beta_1$ and $\beta_2$ can interchange |

- What is the optimal convergence rate by tuning the step size or other parameters?
- Can we prove something similar to the Stein–Rosenberg theorem for Jacobi and GS updates?

**Contributions**  We summarize our main results from §3 and §4 in Table 1 and 2 respectively, with supporting experiments given in §5. We use $\sigma_1$ and $\sigma_n$ to denote the largest and the smallest singular values of matrix $\boldsymbol{E}$ (see equation 2.1), and $\kappa := \sigma_1/\sigma_n$ denotes the condition number. The algorithms will be introduced in §2. Note that we generalize gradient-type algorithms but retain the same names. Table 1 shows that in most cases that we study, whenever Jacobi updates converge, the corresponding GS updates converge as well (usually with a faster rate), but the converse is not true (§3). This extends the well-known Stein–Rosenberg theorem to bilinear games. Furthermore, Table 2 tells us that by generalizing existing gradient algorithms, we can obtain faster convergence rates.

## 2 PRELIMINARIES

In the study of GAN training, bilinear games are often regarded as an important simple example for theoretically analyzing and understanding new algorithms and techniques (e.g. Daskalakis et al., 2018; Gidel et al., 2019a;b; Liang & Stokes, 2019). It captures the difficulty in GAN training and can represent some simple GAN formulations (Arjovsky et al., 2017; Daskalakis et al., 2018; Gidel et al., 2019a; Mescheder et al., 2018). Mathematically, *bilinear* zero-sum games can be formulated as the following min-max problem:

$$\min_{\boldsymbol{x} \in \mathbb{R}^n} \max_{\boldsymbol{y} \in \mathbb{R}^n} \quad \boldsymbol{x}^\top \boldsymbol{E} \boldsymbol{y} + \boldsymbol{b}^\top \boldsymbol{x} + \boldsymbol{c}^\top \boldsymbol{y}. \tag{2.1}$$

The set of all saddle points (see definition in eq. (1.1)) is:

$$\{(\boldsymbol{x}, \boldsymbol{y}) \,|\, \boldsymbol{E}\boldsymbol{y} + \boldsymbol{b} = \boldsymbol{0}, \, \boldsymbol{E}^\top \boldsymbol{x} + \boldsymbol{c} = \boldsymbol{0}\}. \tag{2.2}$$

Throughout, for simplicity we assume $\boldsymbol{E}$ to be invertible, whereas the seemingly general case with non-invertible $\boldsymbol{E}$ is treated in Appendix G. The linear terms are not essential in our analysis and we take $\boldsymbol{b} = \boldsymbol{c} = \boldsymbol{0}$ throughout the paper[1]. In this case, the only saddle point is $(\boldsymbol{0}, \boldsymbol{0})$. For bilinear games, it is well-known that simultaneous gradient descent ascent does not converge (Nemirovski & Yudin, 1983) and other gradient-based algorithms tailored for min-max optimization have been proposed (Korpelevich, 1976; Daskalakis et al., 2018; Gidel et al., 2019a; Mescheder et al., 2017). These iterative algorithms all belong to the class of general linear dynamical systems (LDS, a.k.a.

---

[1]If they are not zero, one can translate $\boldsymbol{x}$ and $\boldsymbol{y}$ to cancel the linear terms, see e.g. Gidel et al. (2019b).

matrix iterative processes). Using state augmentation $\boldsymbol{z}^{(t)} := (\boldsymbol{x}^{(t)}, \boldsymbol{y}^{(t)})$ we define a general $k$-step LDS as follows:

$$\boldsymbol{z}^{(t)} = \sum_{i=1}^{k} \boldsymbol{A}_i \boldsymbol{z}^{(t-i)} + \boldsymbol{d}, \tag{2.3}$$

where the matrices $\boldsymbol{A}_i$ and vector $\boldsymbol{d}$ depend on the gradient algorithm (examples can be found in Appendix C.1). Define the characteristic polynomial, with $\boldsymbol{A}_0 = -\boldsymbol{I}$:

$$p(\lambda) := \det(\sum_{i=0}^{k} \boldsymbol{A}_i \lambda^{k-i}). \tag{2.4}$$

The following well-known result decides when such a $k$-step LDS converges for any initialization:

**Theorem 2.1** (e.g. Gohberg et al. (1982)). *The LDS in eq. (2.3) converges for any initialization $(\boldsymbol{z}^{(0)}, \ldots, \boldsymbol{z}^{(k-1)})$ iff the spectral radius $r := \max\{|\lambda| : p(\lambda) = 0\} < 1$, in which case $\{\boldsymbol{z}^{(t)}\}$ converges linearly with an (asymptotic) exponent $r$.*

Therefore, understanding the bilinear game dynamics reduces to spectral analysis. The (sufficient and necessary) convergence condition reduces to that all roots of $p(\lambda)$ lie in the (open) unit disk, which can be conveniently analyzed through the celebrated Schur's theorem (Schur, 1917):

**Theorem 2.2** (Schur (1917)). *The roots of a real polynomial $p(\lambda) = a_0 \lambda^n + a_1 \lambda^{n-1} + \cdots + a_n$ are within the (open) unit disk of the complex plane iff $\forall k \in \{1, 2, \ldots, n\}$, $\det(\boldsymbol{P}_k \boldsymbol{P}_k^\top - \boldsymbol{Q}_k^\top \boldsymbol{Q}_k) > 0$, where $\boldsymbol{P}_k, \boldsymbol{Q}_k$ are $k \times k$ matrices defined as: $[\boldsymbol{P}_k]_{i,j} = a_{i-j} \mathbf{1}_{i \geq j}$, $[\boldsymbol{Q}_k]_{i,j} = a_{n-i+j} \mathbf{1}_{i \leq j}$.*

In the theorem above, we denoted $\mathbf{1}_S$ as the indicator function of the event $S$, i.e. $\mathbf{1}_S = 1$ if $S$ holds and $\mathbf{1}_S = 0$ otherwise. For a nice summary of related stability tests, see Mansour (2011). We therefore define *Schur stable* polynomials to be those polynomials whose roots all lie within the (open) unit disk of the complex plane. Schur's theorem has the following corollary (proof included in Appendix B.2 for the sake of completeness):

**Corollary 2.1** (e.g. Mansour (2011)). *A real quadratic polynomial $\lambda^2 + a\lambda + b$ is Schur stable iff $b < 1$, $|a| < 1 + b$; A real cubic polynomial $\lambda^3 + a\lambda^2 + b\lambda + c$ is Schur stable iff $|c| < 1$, $|a + c| < 1 + b$, $b - ac < 1 - c^2$; A real quartic polynomial $\lambda^4 + a\lambda^3 + b\lambda^2 + c\lambda + d$ is Schur stable iff $|c - ad| < 1 - d^2$, $|a + c| < b + d + 1$, and $b < (1 + d) + (c - ad)(a - c)/(d - 1)^2$.*

Let us formally define Jacobi and GS updates: Jacobi updates take the form

$$\boldsymbol{x}^{(t)} = T_1(\boldsymbol{x}^{(t-1)}, \boldsymbol{y}^{(t-1)}, \ldots, \boldsymbol{x}^{(t-k)}, \boldsymbol{y}^{(t-k)}), \quad \boldsymbol{y}^{(t)} = T_2(\boldsymbol{x}^{(t-1)}, \boldsymbol{y}^{(t-1)}, \ldots, \boldsymbol{x}^{(t-k)}, \boldsymbol{y}^{(t-k)}),$$

while Gauss–Seidel updates replace $\boldsymbol{x}^{(t-i)}$ with the more recent $\boldsymbol{x}^{(t-i+1)}$ in operator $T_2$, where $T_1, T_2 : \mathbb{R}^{nk} \times \mathbb{R}^{nk} \to \mathbb{R}^n$ can be any update functions. For LDS updates in eq. (2.3) we find a nice relation between the characteristic polynomials of Jacobi and GS updates in Theorem 2.3 (proof in Appendix B.1), which turns out to greatly simplify our subsequent analyses:

**Theorem 2.3** (**Jacobi vs. Gauss–Seidel**). *Let $p(\lambda, \gamma) = \det(\sum_{i=0}^{k}(\gamma \boldsymbol{L}_i + \boldsymbol{U}_i)\lambda^{k-i})$, where $\boldsymbol{A}_i = \boldsymbol{L}_i + \boldsymbol{U}_i$ and $\boldsymbol{L}_i$ is strictly lower block triangular. Then, the characteristic polynomial of Jacobi updates is $p(\lambda, 1)$ while that of Gauss–Seidel updates is $p(\lambda, \lambda)$.*

Compared to the Jacobi update, in some sense the Gauss–Seidel update amounts to *shifting the strictly lower block triangular matrices $\boldsymbol{L}_i$ one step to the left*, as $p(\lambda, \lambda)$ can be rewritten as $\det\left(\sum_{i=0}^{k}(\boldsymbol{L}_{i+1} + \boldsymbol{U}_i)\lambda^{k-i}\right)$, with $\boldsymbol{L}_{k+1} := \mathbf{0}$. This observation will significantly simplify our comparison between Jacobi and Gauss–Seidel updates.

Next, we define some popular gradient algorithms for finding saddle points in the min-max problem

$$\min_{\boldsymbol{x}} \max_{\boldsymbol{y}} f(\boldsymbol{x}, \boldsymbol{y}). \tag{2.5}$$

We present the algorithms for a general (bivariate) function $f$ although our main results will specialize $f$ to the bilinear case in eq. (2.1). Note that we introduced more "step sizes" for our refined analysis, as we find that the enlarged parameter space often contains choices for faster linear convergence (see §4). We only define the Jacobi updates, while the GS counterparts can be easily inferred. We always use $\alpha_1$ and $\alpha_2$ to define step sizes (or learning rates) which are positive.

**Gradient descent (GD)** The generalized GD update has the following form:

$$\boldsymbol{x}^{(t+1)} = \boldsymbol{x}^{(t)} - \alpha_1 \nabla_{\boldsymbol{x}} f(\boldsymbol{x}^{(t)}, \boldsymbol{y}^{(t)}), \qquad \boldsymbol{y}^{(t+1)} = \boldsymbol{y}^{(t)} + \alpha_2 \nabla_{\boldsymbol{y}} f(\boldsymbol{x}^{(t)}, \boldsymbol{y}^{(t)}). \qquad (2.6)$$

When $\alpha_1 = \alpha_2$, the convergence of averaged iterates (a.k.a. Cesari convergence) for convex-concave games is analyzed in (Bruck, 1977; Nemirovski & Yudin, 1978; Nedić & Ozdaglar, 2009). Recent progress on interpreting GD with dynamical systems can be seen in, e.g., Mertikopoulos et al. (2018); Bailey et al. (2019); Bailey & Piliouras (2018).

**Extra-gradient (EG)** We study a generalized version of EG, defined as follows:

$$\boldsymbol{x}^{(t+1/2)} = \boldsymbol{x}^{(t)} - \gamma_1 \nabla_{\boldsymbol{x}} f(\boldsymbol{x}^{(t)}, \boldsymbol{y}^{(t)}), \ \boldsymbol{y}^{(t+1/2)} = \boldsymbol{y}^{(t)} + \gamma_2 \nabla_{\boldsymbol{y}} f(\boldsymbol{x}^{(t)}, \boldsymbol{y}^{(t)}); \qquad (2.7)$$

$$\boldsymbol{x}^{(t+1)} = \boldsymbol{x}^{(t)} - \alpha_1 \nabla_{\boldsymbol{x}} f(\boldsymbol{x}^{(t+1/2)}, \boldsymbol{y}^{(t+1/2)}), \ \boldsymbol{y}^{(t+1)} = \boldsymbol{y}^{(t)} + \alpha_2 \nabla_{\boldsymbol{y}} f(\boldsymbol{x}^{(t+1/2)}, \boldsymbol{y}^{(t+1/2)}). \ (2.8)$$

EG was first proposed in Korpelevich (1976) with the restriction $\alpha_1 = \alpha_2 = \gamma_1 = \gamma_2$, under which linear convergence was proved for bilinear games. Convergence of EG on convex-concave games was analyzed in Nemirovski (2004); Monteiro & Svaiter (2010), and Mertikopoulos et al. (2019) provides convergence guarantees for specific non-convex-non-concave problems. For bilinear games, a slightly more generalized version was proposed in Liang & Stokes (2019) where $\alpha_1 = \alpha_2$, $\gamma_1 = \gamma_2$, with linear convergence proved. For later convenience we define $\beta_1 = \alpha_2 \gamma_1$ and $\beta_2 = \alpha_1 \gamma_2$.

**Optimistic gradient descent (OGD)** We study a generalized version of OGD, defined as follows:

$$\boldsymbol{x}^{(t+1)} = \boldsymbol{x}^{(t)} - \alpha_1 \nabla_{\boldsymbol{x}} f(\boldsymbol{x}^{(t)}, \boldsymbol{y}^{(t)}) + \beta_1 \nabla_{\boldsymbol{x}} f(\boldsymbol{x}^{(t-1)}, \boldsymbol{y}^{(t-1)}), \qquad (2.9)$$

$$\boldsymbol{y}^{(t+1)} = \boldsymbol{y}^{(t)} + \alpha_2 \nabla_{\boldsymbol{y}} f(\boldsymbol{x}^{(t)}, \boldsymbol{y}^{(t)}) - \beta_2 \nabla_{\boldsymbol{y}} f(\boldsymbol{x}^{(t-1)}, \boldsymbol{y}^{(t-1)}). \qquad (2.10)$$

The original version of OGD was given in Popov (1980) with $\alpha_1 = \alpha_2 = 2\beta_1 = 2\beta_2$ and rediscovered in the GAN literature (Daskalakis et al., 2018). Its linear convergence for bilinear games was proved in Liang & Stokes (2019). A slightly more generalized version with $\alpha_1 = \alpha_2$ and $\beta_1 = \beta_2$ was analyzed in Peng et al. (2019); Mokhtari et al. (2019b), again with linear convergence proved. The stochastic case was analyzed in Hsieh et al. (2019).

**Momentum method** Generalized heavy ball method was analyzed in Gidel et al. (2019b):

$$\boldsymbol{x}^{(t+1)} = \boldsymbol{x}^{(t)} - \alpha_1 \nabla_{\boldsymbol{x}} f(\boldsymbol{x}^{(t)}, \boldsymbol{y}^{(t)}) + \beta_1 (\boldsymbol{x}^{(t)} - \boldsymbol{x}^{(t-1)}), \qquad (2.11)$$

$$\boldsymbol{y}^{(t+1)} = \boldsymbol{y}^{(t)} + \alpha_2 \nabla_{\boldsymbol{y}} f(\boldsymbol{x}^{(t)}, \boldsymbol{y}^{(t)}) + \beta_2 (\boldsymbol{y}^{(t)} - \boldsymbol{y}^{(t-1)}). \qquad (2.12)$$

This is a modification of Polyak's heavy ball (HB) (Polyak, 1964), which also motivated Nesterov's accelerated gradient algorithm (NAG) (Nesterov, 1983). Note that for both $\boldsymbol{x}$-update and the $\boldsymbol{y}$-update, we *add* a scale multiple of the successive difference (e.g. proxy of the momentum). For this algorithm our result below improves those obtained in Gidel et al. (2019b), as will be discussed in §3.

**EG and OGD as approximations of proximal point algorithm** It has been observed recently in Mokhtari et al. (2019b) that for convex-concave games, EG ($\alpha_1 = \alpha_2 = \gamma_1 = \gamma_2 = \eta$) and OGD ($\alpha_1/2 = \alpha_2/2 = \beta_1 = \beta_2 = \eta$) can be treated as approximations of the proximal point algorithm (Martinet, 1970; Rockafellar, 1976) when $\eta$ is small. With this result, one can show that EG and OGD converge to saddle points sublinearly for smooth convex-concave games (Mokhtari et al., 2019a). We give a brief introduction of the proximal point algorithm in Appendix A (including a linear convergence result for the slightly generalized version).

The above algorithms, when specialized to a bilinear function $f$ (see eq. (2.1)), can be rewritten as a 1-step or 2-step LDS (see. eq. (2.3)). See Appendix C.1 for details.

## 3 EXACT CONDITIONS

With tools from §2, we formulate necessary and sufficient conditions under which a gradient-based algorithm converges for bilinear games. We sometimes use "J" as a shorthand for Jacobi style updates and "GS" for Gauss–Seidel style updates. For each algorithm, we first write down the characteristic polynomials (see derivation in Appendix C.1) for both Jacobi and GS updates, and present the exact conditions for convergence. Specifically, we show that in many cases the GS convergence regions strictly include the Jacobi convergence regions. The proofs for Theorem 3.1, 3.2, 3.3 and 3.4 can be found in Appendix C.2, C.3, C.4, and C.5, respectively.

**GD**  The characteristic equations can be computed as:

$$\text{J: } (\lambda - 1)^2 + \alpha_1 \alpha_2 \sigma^2 = 0, \text{ GS: } (\lambda - 1)^2 + \alpha_1 \alpha_2 \sigma^2 \lambda = 0. \tag{3.1}$$

**Scaling symmetry**  From section 3 we obtain a scaling symmetry $(\alpha_1, \alpha_2) \to (t\alpha_1, \alpha_2/t)$, with $t > 0$. With this symmetry we can always fix $\alpha_1 = \alpha_2 = \alpha$. This symmetry also holds for EG and momentum. For OGD, the scaling symmetry is slightly different with $(\alpha_1, \beta_1, \alpha_2, \beta_2) \to (t\alpha_1, t\beta_1, \alpha_2/t, \beta_2/t)$, but we can still use this symmetry to fix $\alpha_1 = \alpha_2 = \alpha$.

**Theorem 3.1** (**GD**). *Jacobi GD and Gauss–Seidel GD do not converge. However, Gauss–Seidel GD can have a limit cycle while Jacobi GD always diverges.*

In the constrained case, Mertikopoulos et al. (2018) and Bailey & Piliouras (2018) show that FTRL, a more generalized algorithm of GD, does not converge for polymatrix games. When $\alpha_1 = \alpha_2$, the result of Gauss–Seidel GD has been shown in Bailey et al. (2019).

**EG**  The characteristic equations can be computed as:

$$\text{J: } \qquad (\lambda - 1)^2 + (\beta_1 + \beta_2)\sigma^2(\lambda - 1) + (\alpha_1 \alpha_2 \sigma^2 + \beta_1 \beta_2 \sigma^4) = 0, \tag{3.2}$$
$$\text{GS: } \quad (\lambda - 1)^2 + (\alpha_1 \alpha_2 + \beta_1 + \beta_2)\sigma^2(\lambda - 1) + (\alpha_1 \alpha_2 \sigma^2 + \beta_1 \beta_2 \sigma^4) = 0. \tag{3.3}$$

**Theorem 3.2** (**EG**). *For generalized EG with $\alpha_1 = \alpha_2 = \alpha$ and $\gamma_i = \beta_i/\alpha$, Jacobi and Gauss–Seidel updates achieve linear convergence iff for any singular value $\sigma$ of $\boldsymbol{E}$, we have:*

$$\text{J} : |\beta_1 \sigma^2 + \beta_2 \sigma^2 - 2| < 1 + (1 - \beta_1 \sigma^2)(1 - \beta_2 \sigma^2) + \alpha^2 \sigma^2,$$
$$(1 - \beta_1 \sigma^2)(1 - \beta_2 \sigma^2) + \alpha^2 \sigma^2 < 1, \tag{3.4}$$
$$\text{GS} : |(\beta_1 + \beta_2 + \alpha^2)\sigma^2 - 2| < 1 + (1 - \beta_1 \sigma^2)(1 - \beta_2 \sigma^2),$$
$$(1 - \beta_1 \sigma^2)(1 - \beta_2 \sigma^2) < 1. \tag{3.5}$$

*If $\beta_1 + \beta_2 + \alpha^2 < 2/\sigma_1^2$, the convergence region of GS updates **strictly** include that of Jacobi updates.*

**OGD**  The characteristic equations can be computed as:

$$\text{J: } \quad \lambda^2(\lambda - 1)^2 + (\lambda\alpha_1 - \beta_1)(\lambda\alpha_2 - \beta_2)\sigma^2 = 0, \tag{3.6}$$
$$\text{GS: } \quad \lambda^2(\lambda - 1)^2 + (\lambda\alpha_1 - \beta_1)(\lambda\alpha_2 - \beta_2)\lambda\sigma^2 = 0. \tag{3.7}$$

**Theorem 3.3** (**OGD**). *For generalized OGD with $\alpha_1 = \alpha_2 = \alpha$, Jacobi and Gauss–Seidel updates achieve linear convergence iff for any singular value $\sigma$ of $\boldsymbol{E}$, we have:*

$$\text{J} : \begin{cases} |\beta_1 \beta_2 \sigma^2| < 1, (\alpha - \beta_1)(\alpha - \beta_2) > 0, 4 + (\alpha + \beta_1)(\alpha + \beta_2)\sigma^2 > 0, \\ \alpha^2 \left(\beta_1^2 \sigma^2 + 1\right)\left(\beta_2^2 \sigma^2 + 1\right) < (\beta_1 \beta_2 \sigma^2 + 1)(2\alpha(\beta_1 + \beta_2) + \beta_1 \beta_2(\beta_1 \beta_2 \sigma^2 - 3)); \end{cases} \tag{3.8}$$

$$\text{GS} : \begin{cases} (\alpha - \beta_1)(\alpha - \beta_2) > 0, (\alpha + \beta_1)(\alpha + \beta_2)\sigma^2 < 4, \\ (\alpha\beta_1 \sigma^2 + 1)(\alpha\beta_2 \sigma^2 + 1) > (1 + \beta_1 \beta_2 \sigma^2)^2. \end{cases} \tag{3.9}$$

*The convergence region of GS updates **strictly** include that of Jacobi updates.*

**Momentum**  The characteristic equations can be computed as:

$$\text{J: } (\lambda - 1)^2(\lambda - \beta_1)(\lambda - \beta_2) + \alpha_1 \alpha_2 \sigma^2 \lambda^2 = 0, \tag{3.10}$$
$$\text{GS: } (\lambda - 1)^2(\lambda - \beta_1)(\lambda - \beta_2) + \alpha_1 \alpha_2 \sigma^2 \lambda^3 = 0. \tag{3.11}$$

**Theorem 3.4** (**momentum**). *For the generalized momentum method with $\alpha_1 = \alpha_2 = \alpha$, the Jacobi updates never converge, while the GS updates converge iff for any singular value $\sigma$ of $\boldsymbol{E}$, we have:*

$$|\beta_1 \beta_2| < 1, |-\alpha^2 \sigma^2 + \beta_1 + \beta_2 + 2| < \beta_1 \beta_2 + 3, 4(\beta_1 + 1)(\beta_2 + 1) > \alpha^2 \sigma^2,$$
$$\alpha^2 \sigma^2 \beta_1 \beta_2 < (1 - \beta_1 \beta_2)(2\beta_1 \beta_2 - \beta_1 - \beta_2). \tag{3.12}$$

*This condition implies that at least one of $\beta_1, \beta_2$ is **negative**.*

Prior to our work, only sufficient conditions for linear convergence were given for the usual EG and OGD; see §2 above. For the momentum method, our result improves upon Gidel et al. (2019b) where they only considered specific cases of parameters. For example, they only considered $\beta_1 = \beta_2 \geq -1/16$ for Jacobi momentum (but with explicit rate of divergence), and $\beta_1 = -1/2$, $\beta_2 = 0$ for GS momentum (with convergence rate). Our Theorem 3.4 gives a more complete picture and formally justifies the necessity of negative momentum.

In the theorems above, we used the term "convergence region" to denote a subset of the parameter space (with parameters $\alpha$, $\beta$ or $\gamma$) where the algorithm converges. Our result shares similarity with the celebrated Stein–Rosenberg theorem (Stein & Rosenberg, 1948), which only applies to solving linear systems with non-negative matrices (if one were to apply it to our case, the matrix $\boldsymbol{S}$ in eq. (F.1) in Appendix F needs to have non-zero diagonal entries, which is not possible). In this sense, our results extend the Stein–Rosenberg theorem to cover nontrivial bilinear games.

## 4 OPTIMAL EXPONENTS OF LINEAR CONVERGENCE

In this section we study the optimal convergence rates of EG and OGD. We define the exponent of linear convergence as $r = \lim_{t \to \infty} ||\boldsymbol{z}^{(t)}||/||\boldsymbol{z}^{(t-1)}||$ which is the same as the spectral radius. For ease of presentation we fix $\alpha_1 = \alpha_2 = \alpha > 0$ (using scaling symmetry) and we use $r_*$ to denote the optimal exponent of linear convergence (achieved by tuning the parameters $\alpha, \beta, \gamma$). Our results show that by generalizing gradient algorithms one can obtain better convergence rates.

**Theorem 4.1** (**EG optimal**). *Both Jacobi and GS EG achieve the optimal exponent of linear convergence* $r_* = (\kappa^2 - 1)/(\kappa^2 + 1)$ *at* $\alpha \to 0$ *and* $\beta_1 = \beta_2 = 2/(\sigma_1^2 + \sigma_n^2)$*. As* $\kappa \to \infty$*,* $r_* \to 1 - 2/\kappa^2$*.*

Note that we defined $\beta_i = \gamma_i \alpha$ in Section 2. In other words, we are taking very large extra-gradient steps ($\gamma_i \to \infty$) and very small gradient steps ($\alpha \to 0$).

**Theorem 4.2** (**OGD optimal**). *For Jacobi OGD with* $\beta_1 = \beta_2 = \beta$*, to achieve the optimal exponent of linear convergence, we must have* $\alpha \leq 2\beta$*. For the original OGD with* $\alpha = 2\beta$*, the optimal exponent of linear convergence* $r_*$ *satisfies*

$$r_*^2 = \frac{1}{2} + \frac{1}{4\sqrt{2}\sigma_1^2}\sqrt{(\sigma_1^2 - \sigma_n^2)(5\sigma_1^2 - \sigma_n^2 + \sqrt{(\sigma_1^2 - \sigma_n^2)(9\sigma_1^2 - \sigma_n^2)})}, \ at \tag{4.1}$$

$$\beta_* = \frac{1}{4\sqrt{2}}\sqrt{\frac{3\sigma_1^4 - (\sigma_1^2 - \sigma_n^2)^{3/2}\sqrt{9\sigma_1^2 - \sigma_n^2} + 6\sigma_1^2\sigma_n^2 - \sigma_n^4}{\sigma_1^4\sigma_n^2}}. \tag{4.2}$$

*If* $\kappa \to \infty$*,* $r_* \sim 1 - 1/(6\kappa^2)$*. For GS OGD with* $\beta_2 = 0$*, the optimal exponent of convergence is* $r_* = \sqrt{(\kappa^2 - 1)/(\kappa^2 + 1)}$*, at* $\alpha = \sqrt{2}/\sigma_1$ *and* $\beta_1 = \sqrt{2}\sigma_1/(\sigma_1^2 + \sigma_n^2)$*. If* $\kappa \to \infty$*,* $r_* \sim 1 - 1/\kappa^2$*.*

**Remark** The original OGD (Popov, 1980; Daskalakis et al., 2018) with $\alpha = 2\beta$ may not always be optimal. For example, take one-dimensional bilinear game and $\sigma = 1$, and denote the spectral radius given $\alpha, \beta$ as $r(\alpha, \beta)$. If we fix $\alpha = 1/2$, by numerically solving section 3 we have

$$r(1/2, 1/4) \approx 0.966, \ r(1/2, 1/3) \approx 0.956, \tag{4.3}$$

i.e, $\alpha = 1/2$, $\beta = 1/3$ is a better choice than $\alpha = 2\beta = 1/2$.

**Numerical method** We provide a numerical method for finding the optimal exponent of linear convergence, by realizing that the *unit* disk in Theorem 2.2 is not special. Let us call a polynomial to be $r$-Schur stable if all of its roots lie within an (open) disk of radius $r$ in the complex plane. We can scale the polynomial with the following lemma:

**Lemma 4.1.** *A polynomial* $p(\lambda)$ *is* $r$-Schur stable iff $p(r\lambda)$ *is Schur stable.*

With the lemma above, one can rescale the Schur conditions and find the convergence region where the exponent of linear convergence is at most $r$ ($r < 1$). A simple binary search would allow one to find a better and better convergence region. See details in Appendix D.3.

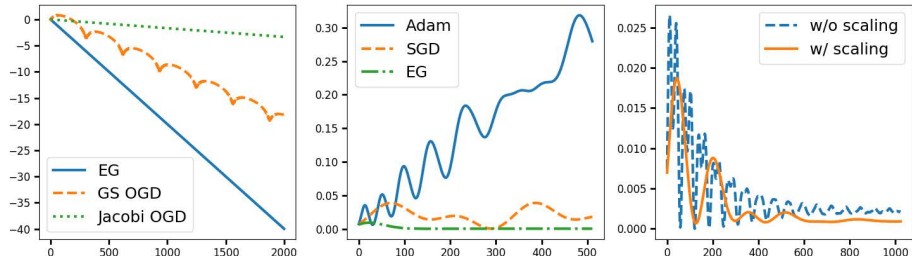

Figure 1: **Left:** linear convergence of optimal EG, Jacobi OGD, Gauss–Seidel OGD in a bilinear game with the log distance; **Middle:** comparison among Adam, SGD and EG in learning the mean of a Gaussian with WGAN with the squared distance; **Right:** Comparison between EG with ($\alpha = 0.02$, $\gamma = 2.0$) and without scaling ($\alpha = \gamma = 0.2$). We use the squared distance.

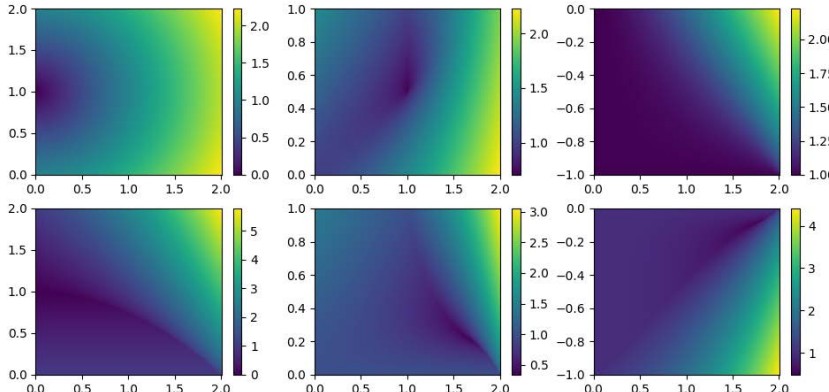

Figure 2: Heat maps of the spectral radii of different algorithms. We take $\sigma = 1$ for convenience. The horizontal axis is $\alpha$ and the vertical axis is $\beta$. **Top row:** Jacobi updates; **Bottom row:** Gauss–Seidel updates. **Columns** (left to right): EG; OGD; momentum. If the spectral radius is strictly less than one, it means that our algorithm converges. In each column, the Jacobi convergence region is contained in the GS convergence region (for EG we need an additional assumption, see Theorem 3.2).

## 5 EXPERIMENTS

**Bilinear game**    We run experiments on a simple bilinear game and choose the optimal parameters as suggested in Theorem 4.1 and 4.2. The results are shown in the left panel of Figure 1, which confirms the predicted linear rates.

**Density plots**    We show the density plots (heat maps) of the spectral radii in Figure 2. We make plots for EG, OGD and momentum with both Jacobi and GS updates. These plots are made when $\beta_1 = \beta_2 = \beta$ and they agree with our theorems in §3.

**Wasserstein GAN**    As in Daskalakis et al. (2018), we consider a WGAN (Arjovsky et al., 2017) that learns the mean of a Gaussian:

$$\min_{\phi} \max_{\theta} f(\phi, \theta) := \mathbb{E}_{\boldsymbol{x} \sim \mathcal{N}(\boldsymbol{v}, \sigma^2 \boldsymbol{I})}[s(\boldsymbol{\theta}^\top \boldsymbol{x})] - \mathbb{E}_{\boldsymbol{z} \sim \mathcal{N}(\boldsymbol{0}, \sigma^2 \boldsymbol{I})}[s(\boldsymbol{\theta}^\top (\boldsymbol{z} + \boldsymbol{\phi}))], \qquad (5.1)$$

where $s(x)$ is the sigmoid function. It can be shown that near the saddle point $(\boldsymbol{\theta}^*, \boldsymbol{\phi}^*) = (\boldsymbol{0}, \boldsymbol{v})$ the min-max optimization can be treated as a bilinear game (Appendix E.1). With GS updates, we find that Adam diverges, SGD goes around a limit cycle, and EG converges, as shown in the middle panel of Figure 1. We can see that Adam does not behave well even in this simple task of learning a single two-dimensional Gaussian with GAN.

Our next experiment shows that generalized algorithms may have an advantage over traditional ones. Inspired by Theorem 4.1, we compare the convergence of two EGs with the same parameter $\beta = \alpha\gamma$, and find that with scaling, EG has better convergence, as shown in the right panel of Figure 1. Finally,

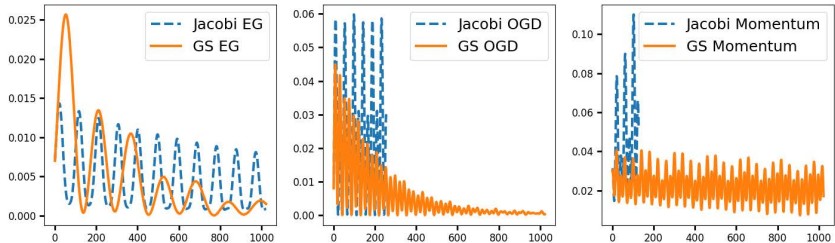

Figure 3: Jacobi vs. GS updates. **y-axis:** Squared distance $||\phi - v||^2$. **x-axis:** Number of epochs. **Left:** EG with $\gamma = 0.2, \alpha = 0.02$; **Middle:** OGD with $\alpha = 0.2$, $\beta_1 = 0.1$, $\beta_2 = 0$; **Right:** Momentum with $\alpha = 0.08$, $\beta = -0.1$. We plot only a few epochs for Jacobi if it does not converge.

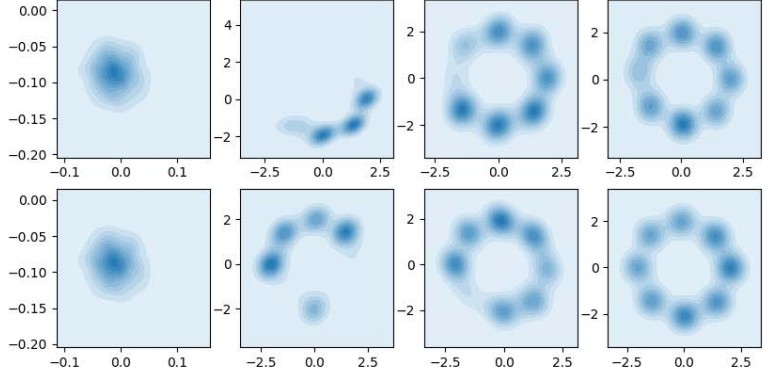

Figure 4: Test samples from the generator network trained with stochastic GD (step size $\alpha = 0.01$). **Top row:** Jacobi updates; **Bottom row:** Gauss–Seidel updates. **Columns**: epoch 0, 10, 15, 20.

we compare Jacobi updates with GS updates. In Figure 3, we can see that GS updates converge even if the corresponding Jacobi updates do not.

**Mixtures of Gaussians (GMMs)** Our last experiment is on learning GMMs with a vanilla GAN (Goodfellow et al., 2014) that does not directly fall into our analysis. We choose a 3-hidden layer ReLU network for both the generator and the discriminator, and each hidden layer has 256 units. We find that for GD and OGD, Jacobi style updates converge more slowly than GS updates, and whenever Jacobi updates converge, the corresponding GS updates converges as well. These comparisons can be found in Figure 4 and 5, which implies the possibility of extending our results to non-bilinear games. Interestingly, we observe that even Jacobi GD converges on this example. We provide additional comparison between the Jacobi and GS updates of Adam (Kingma & Ba, 2015) in Appendix E.2.

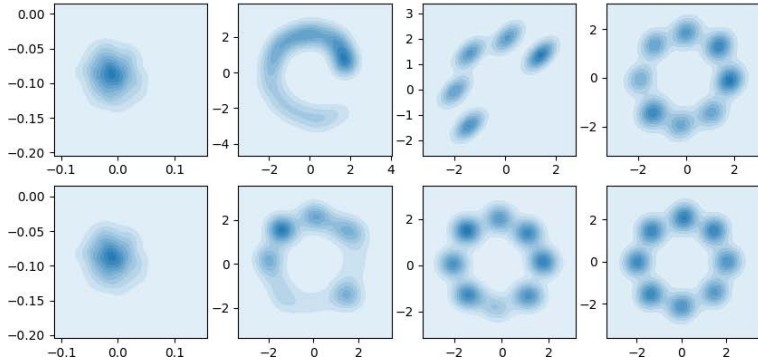

Figure 5: Test samples from the generator network trained with stochastic OGD ($\alpha = 2\beta = 0.02$). **Top row:** Jacobi updates; **Bottom row:** Gauss–Seidel updates. **Columns**: epoch 0, 10, 60, 100.

## 6 CONCLUSIONS

In this work we focus on the convergence behaviour of gradient-based algorithms for solving bilinear games. By drawing a connection to discrete linear dynamical systems (§2) and using Schur's theorem, we provide necessary and sufficient conditions for a variety of gradient algorithms, for both simultaneous (Jacobi) and alternating (Gauss–Seidel) updates. Our results show that Gauss–Seidel updates converge more easily than Jacobi updates. Furthermore, we find the optimal exponents of linear convergence for EG and OGD, and provide a numerical method for searching that exponent. We performed a number of experiments to validate our theoretical findings and suggest further analysis.

There are many future directions to explore. For example, our preliminary experiments on GANs suggest that similar (local) results might be obtained for more general games. Indeed, the local convergence behaviour of min-max nonlinear optimization can be studied through analyzing the spectrum of the Jacobian matrix of the update operator (see, e.g., Nagarajan & Kolter (2017); Gidel et al. (2019b)). We believe our framework that draws the connection to linear discrete dynamic systems and Schur's theorem is a powerful machinery that can be applied in such problems and beyond. It would be interesting to generalize our results to the constrained case (even for bilinear games), as studied in Daskalakis & Panageas (2019); Carmon et al. (2019). Extending our results to account for stochastic noise (as empirically tested in our experiments) is another interesting direction, with results in Gidel et al. (2019a); Hsieh et al. (2019).

## ACKNOWLEDGEMENTS

We would like to thank Argyrios Deligkas, Sarath Pattathil and Georgios Piliouras for pointing out several related references. GZ is supported by David R. Cheriton Scholarship. We gratefully acknowledge funding support from NSERC and the Waterloo-Huawei Joint Innovation Lab.

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

## A PROXIMAL POINT (PP) ALGORITHM

PP was originally proposed by Martinet (1970) with $\alpha_1 = \alpha_2$ and then carefully studied by Rockafellar (1976). The linear convergence for bilinear games was also proved in the same reference. Note that we do not consider Gauss–Seidel PP since we do not get a meaningful solution after a shift of steps[2].

$$\boldsymbol{x}^{(t+1)} = \boldsymbol{x}^{(t)} - \alpha_1 \nabla_{\boldsymbol{x}} f(\boldsymbol{x}^{(t+1)}, \boldsymbol{y}^{(t+1)}),\ \boldsymbol{y}^{(t+1)} = \boldsymbol{y}^{(t)} + \alpha_2 \nabla_{\boldsymbol{y}} f(\boldsymbol{x}^{(t+1)}, \boldsymbol{y}^{(t+1)}), \qquad \text{(A.1)}$$

where $\boldsymbol{x}^{(t+1)}$ and $\boldsymbol{y}^{(t+1)}$ are given implicitly by solving the equations above. For bilinear games, one can derive that:

$$\boldsymbol{z}^{(t+1)} = \begin{bmatrix} \boldsymbol{I} & \alpha_1 \boldsymbol{E} \\ -\alpha_2 \boldsymbol{E}^\top & \boldsymbol{I} \end{bmatrix}^{-1} \boldsymbol{z}^{(t)}. \qquad \text{(A.2)}$$

We can compute the exact form of the inverse matrix, but perhaps an easier way is just to compute the spectrum of the original matrix (the same as Jacobi GD except that we flip the signs of $\alpha_i$) and perform $\lambda \to 1/\lambda$. Using the fact that the eigenvalues of a matrix are reciprocals of the eigenvalues of its inverse, the characteristic equation is:

$$(1/\lambda - 1)^2 + \alpha_1 \alpha_2 \sigma^2 = 0. \qquad \text{(A.3)}$$

With the scaling symmetry $(\alpha_1, \alpha_2) \to (t\alpha_1, \alpha_2/t)$, we can take $\alpha_1 = \alpha_2 = \alpha > 0$. With the notations in Corollary 2.1, we have $a = -2/(1 + \alpha^2 \sigma^2)$ and $b = 1/(1 + \alpha^2 \sigma^2)$, and it is easy to check $|a| < 1 + b$ and $b < 1$ are always satisfied, which means linear convergence is always guaranteed. Hence, we have the following theorem:

**Theorem A.1.** *For bilinear games, the proximal point algorithm always converges linearly.*

Although the proximal point algorithm behaves well, it is rarely used in practice since it is an implicit method, i.e., one needs to solve $(\boldsymbol{x}^{(t+1)}, \boldsymbol{y}^{(t+1)})$ from equation A.1.

## B PROOFS IN SECTION 2

### B.1 PROOF OF THEOREM 2.3

In this section we apply Theorem 2.1 to prove Theorem 2.3, an interesting connection between Jacobi and Gauss–Seidel updates:

**Theorem 2.3** (**Jacobi vs. Gauss–Seidel**)**.** *Let $p(\lambda, \gamma) = \det(\sum_{i=0}^{k} (\gamma \boldsymbol{L}_i + \boldsymbol{U}_i) \lambda^{k-i})$, where $\boldsymbol{A}_i = \boldsymbol{L}_i + \boldsymbol{U}_i$ and $\boldsymbol{L}_i$ is strictly lower block triangular. Then, the characteristic polynomial of Jacobi updates is $p(\lambda, 1)$ while that of Gauss–Seidel updates is $p(\lambda, \lambda)$.*

Let us first consider the *block* linear iterative process in the sense of Jacobi (i.e., all blocks are updated *simultaneously*):

$$\boldsymbol{z}^{(t)} = \begin{bmatrix} \boldsymbol{z}_1^{(t)} \\ \vdots \\ \boldsymbol{z}_b^{(t)} \end{bmatrix} = \sum_{i=1}^{k} \boldsymbol{A}_i \begin{bmatrix} \boldsymbol{z}_1^{(t-i)} \\ \vdots \\ \boldsymbol{z}_b^{(t-i)} \end{bmatrix} = \sum_{i=1}^{k} \left[ \sum_{j=1}^{l-1} \boldsymbol{A}_{i,j} \boldsymbol{z}_j^{(t-i)} + \sum_{j=l}^{b} \boldsymbol{A}_{i,j} \boldsymbol{z}_j^{(t-i)} \right] + \boldsymbol{d}, \qquad \text{(B.1)}$$

where $\boldsymbol{A}_{i,j}$ is the $j$-th column block of $\boldsymbol{A}_i$. For each matrix $\boldsymbol{A}_i$, we decompose it into the sum

$$\boldsymbol{A}_i = \boldsymbol{L}_i + \boldsymbol{U}_i, \qquad \text{(B.2)}$$

where $\boldsymbol{L}_i$ is the strictly lower *block* triangular part and $\boldsymbol{U}_i$ is the upper (including diagonal) *block* triangular part. Theorem 2.1 indicates that the convergence behaviour of equation B.1 is governed by the largest modulus of the roots of the characteristic polynomial:

$$\det\left( -\lambda^k \boldsymbol{I} + \sum_{i=1}^{k} \boldsymbol{A}_i \lambda^{k-i} \right) = \det\left( -\lambda^k \boldsymbol{I} + \sum_{i=1}^{k} (\boldsymbol{L}_i + \boldsymbol{U}_i) \lambda^{k-i} \right). \qquad \text{(B.3)}$$

---

[2]If one uses inverse operators this is in principle doable.

Alternatively, we can also consider the updates in the sense of Gauss–Seidel (i.e., blocks are updated *sequentially*):

$$z_l^{(t)} = \sum_{i=1}^{k} \left[ \sum_{j=1}^{l-1} A_{i,j} z_j^{(t-i+1)} + \sum_{j=l}^{b} A_{i,j} z_j^{(t-i)} \right]_l + d_l, \quad l = 1, \dots, b. \tag{B.4}$$

We can rewrite the Gauss–Seidel update elegantly[3] as:

$$(I - L_1) z^{(t)} = \sum_{i=1}^{k} (L_{i+1} + U_i) z^{(t-i)} + d, \tag{B.5}$$

i.e.,

$$z^{(t)} = \sum_{i=1}^{k} (I - L_1)^{-1} (L_{i+1} + U_i) z^{(t-i)} + (I - L_1)^{-1} d, \tag{B.6}$$

where $L_{k+1} := 0$. Applying Theorem 2.1 again we know the convergence behaviour of the Gauss–Seidel update is governed by the largest modulus of roots of the characteristic polynomial:

$$\det\left( -\lambda^k I + \sum_{i=1}^{k} (I - L_1)^{-1} (L_{i+1} + U_i) \lambda^{k-i} \right) \tag{B.7}$$

$$= \det\left( (I - L_1)^{-1} \left( -\lambda^k I + \lambda^k L_1 + \sum_{i=1}^{k} (L_{i+1} + U_i) \lambda^{k-i} \right) \right) \tag{B.8}$$

$$= \det(I - L_1)^{-1} \cdot \det\left( \sum_{i=0}^{k} (\lambda L_i + U_i) \lambda^{k-i} \right) \tag{B.9}$$

Note that $A_0 = -I$ and the factor $\det(I - L_1)^{-1}$ can be discarded since multiplying a characteristic polynomial by a non-zero constant factor does not change its roots.

## B.2 PROOF OF COROLLARY 2.1

**Corollary 2.1** (e.g. Mansour (2011))**.** *A real quadratic polynomial $\lambda^2 + a\lambda + b$ is Schur stable iff $b < 1$, $|a| < 1 + b$; A real cubic polynomial $\lambda^3 + a\lambda^2 + b\lambda + c$ is Schur stable iff $|c| < 1$, $|a + c| < 1 + b$, $b - ac < 1 - c^2$; A real quartic polynomial $\lambda^4 + a\lambda^3 + b\lambda^2 + c\lambda + d$ is Schur stable iff $|c - ad| < 1 - d^2$, $|a + c| < b + d + 1$, and $b < (1 + d) + (c - ad)(a - c)/(d - 1)^2$.*

*Proof.* It suffices to prove the result for quartic polynomials. We write down the matrices:

$$P_1 = [1], \; Q_1 = [d], \tag{B.10}$$

$$P_2 = \begin{bmatrix} 1 & 0 \\ a & 1 \end{bmatrix}, \; Q_2 = \begin{bmatrix} d & c \\ 0 & d \end{bmatrix}, \tag{B.11}$$

$$P_3 = \begin{bmatrix} 1 & 0 & 0 \\ a & 1 & 0 \\ b & a & 1 \end{bmatrix}, \; Q_3 = \begin{bmatrix} d & c & b \\ 0 & d & c \\ 0 & 0 & d \end{bmatrix}, \tag{B.12}$$

$$P_4 = \begin{bmatrix} 1 & 0 & 0 & 0 \\ a & 1 & 0 & 0 \\ b & a & 1 & 0 \\ c & b & a & 0 \end{bmatrix}, \; Q_4 = \begin{bmatrix} d & c & b & a \\ 0 & d & c & b \\ 0 & 0 & d & c \\ 0 & 0 & 0 & d \end{bmatrix}. \tag{B.13}$$

We require $\det(P_k P_k^\top - Q_k^\top Q_k) =: \delta_k > 0$, for $k = 1, 2, 3, 4$. If $k = 1$, we have $1 - d^2 > 0$, namely, $|d| < 1$. $\delta_2 > 0$ reduces to $(c - ad)^2 < (1 - d^2)^2$ and thus $|c - ad| < 1 - d^2$ due to the first condition. $\delta_4 > 0$ simplifies to:

$$-((a + c)^2 - (b + d + 1)^2)((b - d - 1)(d - 1)^2 - (a - c)(c - ad))^2 < 0, \tag{B.14}$$

---

[3] This is well-known when $k = 1$, see e.g. Saad (2003).

which yields $|a + c| < |b + d + 1|$. Finally, $\delta_3 > 0$ reduces to:

$$((b - d - 1)(d - 1)^2 - (a - c)(c - ad))((d^2 - 1)(b + d + 1) + (c - ad)(a + c)) > 0. \quad \text{(B.15)}$$

Denote $p(\lambda) := \lambda^4 + a\lambda^3 + b\lambda^2 + c\lambda + d$, we must have $p(1) > 0$ and $p(-1) > 0$, as otherwise there is a real root $\lambda_0$ with $|\lambda_0| \geq 1$. Hence we obtain $b + d + 1 > |a + c| > 0$. Also, from $|c - ad| < 1 - d^2$, we know that:

$$|c - ad| \cdot |a + c| < |b + d + 1|(1 - d^2) = (b + d + 1)(1 - d^2). \quad \text{(B.16)}$$

So, the second factor in B.15 is negative and the positivity of the first factor reduces to:

$$b < (1 + d) + \frac{(c - ad)(a - c)}{(d - 1)^2}. \quad \text{(B.17)}$$

To obtain the Schur condition for cubic polynomials, we take $d = 0$, and the quartic Schur condition becomes:

$$|c| < 1, \; |a + c| < b + 1, \; b - ac < 1 - c^2. \quad \text{(B.18)}$$

To obtain the Schur condition for quadratic polynomials, we take $c = 0$ in the above and write:

$$b < 1, \; |a| < 1 + b. \quad \text{(B.19)}$$

The proof is now complete. □

## C    PROOFS IN SECTION 3

Some of the following proofs in Appendix C.4 and C.5 rely on Mathematica code (mostly with the built-in function `Reduce`) but in principle the code can be verified manually using cylindrical algebraic decomposition.[4]

### C.1    DERIVATION OF CHARACTERISTIC POLYNOMIALS

In this appendix, we derive the exact forms of LDSs (eq. (2.3)) and the characteristic polynomials for all gradient-based methods introduced in §2, with eq. (2.4). The following lemma is well-known and easy to verify using Schur's complement:

**Lemma C.1.** *Given $M \in \mathbb{R}^{2n \times 2n}$, $A \in \mathbb{R}^{n \times n}$ and*

$$M = \begin{bmatrix} A & B \\ C & D \end{bmatrix}. \quad \text{(C.1)}$$

*If $C$ and $D$ commute, then $\det M = \det(AD - BC)$.*

**Gradient descent**    From equation 2.6 the update equation of Jacobi GD can be derived as:

$$z^{(t+1)} = \begin{bmatrix} I & -\alpha_1 E \\ \alpha_2 E^\top & I \end{bmatrix} z^{(t)}, \quad \text{(C.2)}$$

and with Lemma C.1, we compute the characteristic polynomial as in eq. (2.4):

$$\det \begin{bmatrix} (\lambda - 1)I & \alpha_1 E \\ -\alpha_2 E^\top & (\lambda - 1)I \end{bmatrix} = \det[(\lambda - 1)^2 I + \alpha_1 \alpha_2 E E^\top], \quad \text{(C.3)}$$

With spectral decomposition we obtain equation 3.1. Taking $\alpha_2 \to \lambda \alpha_2$ and with Theorem 2.3 we obtain the corresponding GS updates. Therefore, the characteristic polynomials for GD are:

$$\text{J: } (\lambda - 1)^2 + \alpha_1 \alpha_2 \sigma^2 = 0, \; \text{GS: } (\lambda - 1)^2 + \alpha_1 \alpha_2 \sigma^2 \lambda = 0. \quad \text{(C.4)}$$

---

[4]See the online Mathematica documentation.

**Extra-gradient** From eq. (2.7) and eq. (2.8), the update of Jacobi EG is:

$$\boldsymbol{z}^{(t+1)} = \begin{bmatrix} \boldsymbol{I} - \beta_2 \boldsymbol{E}\boldsymbol{E}^\top & -\alpha_1 \boldsymbol{E} \\ \alpha_2 \boldsymbol{E}^\top & \boldsymbol{I} - \beta_1 \boldsymbol{E}^\top \boldsymbol{E} \end{bmatrix} \boldsymbol{z}^{(t)}, \tag{C.5}$$

the characteristic polynomial is:

$$\det \begin{bmatrix} (\lambda-1)\boldsymbol{I} + \beta_2 \boldsymbol{E}\boldsymbol{E}^\top & \alpha_1 \boldsymbol{E} \\ -\alpha_2 \boldsymbol{E}^\top & (\lambda-1)\boldsymbol{I} + \beta_1 \boldsymbol{E}^\top \boldsymbol{E} \end{bmatrix}. \tag{C.6}$$

Since we assumed $\alpha_2 > 0$, we can left multiply the second row by $\beta_2 \boldsymbol{E}/\alpha_2$ and add it to the first row. Hence, we obtain:

$$\det \begin{bmatrix} (\lambda-1)\boldsymbol{I} & \alpha_1 \boldsymbol{E} + (\lambda-1)\beta_2 \boldsymbol{E}/\alpha_2 + \beta_1 \beta_2 \boldsymbol{E}\boldsymbol{E}^\top \boldsymbol{E}/\alpha_2 \\ -\alpha_2 \boldsymbol{E}^\top & (\lambda-1)\boldsymbol{I} + \beta_1 \boldsymbol{E}^\top \boldsymbol{E} \end{bmatrix}. \tag{C.7}$$

With Lemma C.1 the equation above becomes:

$$\det[(\lambda-1)^2 \boldsymbol{I} + (\beta_1 + \beta_2)\boldsymbol{E}^\top \boldsymbol{E}(\lambda-1) + (\alpha_1 \alpha_2 \boldsymbol{E}^\top \boldsymbol{E} + \beta_1 \beta_2 \boldsymbol{E}^\top \boldsymbol{E}\boldsymbol{E}^\top \boldsymbol{E})], \tag{C.8}$$

which simplifies to equation 3.2 with spectral decomposition. Note that to obtain the GS polynomial, we simply take $\alpha_2 \to \lambda\alpha_2$ in the Jacobi polynomial as shown in Theorem 2.3. For the ease of reading we copy the characteristic equations for generalized EG:

$$\text{J: } (\lambda-1)^2 + (\beta_1 + \beta_2)\sigma^2(\lambda-1) + (\alpha_1 \alpha_2 \sigma^2 + \beta_1 \beta_2 \sigma^4) = 0, \tag{C.9}$$

$$\text{GS: } (\lambda-1)^2 + (\alpha_1 \alpha_2 + \beta_1 + \beta_2)\sigma^2(\lambda-1) + (\alpha_1 \alpha_2 \sigma^2 + \beta_1 \beta_2 \sigma^4) = 0. \tag{C.10}$$

**Optimistic gradient descent** We can compute the LDS for OGD with eq. (2.9) and eq. (2.10):

$$\boldsymbol{z}^{(t+2)} = \begin{bmatrix} \boldsymbol{I} & -\alpha_1 \boldsymbol{E} \\ \alpha_2 \boldsymbol{E}^\top & \boldsymbol{I} \end{bmatrix} \boldsymbol{z}^{(t+1)} + \begin{bmatrix} \boldsymbol{0} & \beta_1 \boldsymbol{E} \\ -\beta_2 \boldsymbol{E}^\top & \boldsymbol{0} \end{bmatrix} \boldsymbol{z}^{(t)}, \tag{C.11}$$

With eq. (2.4), the characteristic polynomial for Jacobi OGD is

$$\det \begin{bmatrix} (\lambda^2 - \lambda)\boldsymbol{I} & (\lambda\alpha_1 - \beta_1)\boldsymbol{E} \\ (-\lambda\alpha_2 + \beta_2)\boldsymbol{E}^\top & (\lambda^2 - \lambda)\boldsymbol{I} \end{bmatrix}. \tag{C.12}$$

Taking the determinant and with Lemma C.1 we obtain equation 3.6. The characteristic polynomial for GS updates in equation 3.7 can be subsequently derived with Theorem 2.3, by taking $(\alpha_2, \beta_2) \to (\lambda\alpha_2, \lambda\beta_2)$. For the ease of reading we copy the characteristic polynomials from the main text as:

$$\text{J: } \lambda^2(\lambda-1)^2 + (\lambda\alpha_1 - \beta_1)(\lambda\alpha_2 - \beta_2)\sigma^2 = 0, \tag{C.13}$$

$$\text{GS: } \lambda^2(\lambda-1)^2 + (\lambda\alpha_1 - \beta_1)(\lambda\alpha_2 - \beta_2)\lambda\sigma^2 = 0. \tag{C.14}$$

**Momentum method** With eq. (2.11) and eq. (2.12), the LDS for the momentum method is:

$$\boldsymbol{z}^{(t+2)} = \begin{bmatrix} (1+\beta_1)\boldsymbol{I} & -\alpha_1 \boldsymbol{E} \\ \alpha_2 \boldsymbol{E}^\top & (1+\beta_2)\boldsymbol{I} \end{bmatrix} \boldsymbol{z}^{(t+1)} + \begin{bmatrix} -\beta_1 \boldsymbol{I} & \boldsymbol{0} \\ \boldsymbol{0} & -\beta_2 \boldsymbol{I} \end{bmatrix} \boldsymbol{z}^{(t)}, \tag{C.15}$$

From eq. (2.4), the characteristic polynomial for Jacobi momentum is

$$\det \begin{bmatrix} (\lambda^2 - \lambda(1+\beta_1) + \beta_1)\boldsymbol{I} & \lambda\alpha_1 \boldsymbol{E} \\ -\lambda\alpha_2 \boldsymbol{E}^\top & (\lambda^2 - \lambda(1+\beta_2) + \beta_2)\boldsymbol{I} \end{bmatrix}. \tag{C.16}$$

Taking the determinant and with Lemma C.1 we obtain equation 3.10, while equation 3.11 can be derived with Theorem 2.3, by taking $\alpha_2 \to \lambda\alpha_2$. For the ease of reading we copy the characteristic polynomials from the main text as:

$$\text{J: } (\lambda-1)^2(\lambda-\beta_1)(\lambda-\beta_2) + \alpha_1 \alpha_2 \sigma^2 \lambda^2 = 0, \tag{C.17}$$

$$\text{GS: } (\lambda-1)^2(\lambda-\beta_1)(\lambda-\beta_2) + \alpha_1 \alpha_2 \sigma^2 \lambda^3 = 0. \tag{C.18}$$

## C.2 PROOF OF THEOREM 3.1: SCHUR CONDITIONS OF GD

**Theorem 3.1 (GD).** *Jacobi GD and Gauss–Seidel GD do not converge. However, Gauss–Seidel GD can have a limit cycle while Jacobi GD always diverges.*

*Proof.* With the notations in Corollary 2.1, for Jacobi GD, $b = 1 + \alpha^2\sigma^2 > 1$. For Gauss–Seidel GD, $b = 1$. The Schur conditions are violated. □

## C.3 Proof of Theorem 3.2: Schur conditions of EG

**Theorem 3.2** (EG). *For generalized EG with $\alpha_1 = \alpha_2 = \alpha$ and $\gamma_i = \beta_i/\alpha$, Jacobi and Gauss–Seidel updates achieve linear convergence iff for any singular value $\sigma$ of $E$, we have:*

$$\text{J} : |\beta_1\sigma^2 + \beta_2\sigma^2 - 2| < 1 + (1 - \beta_1\sigma^2)(1 - \beta_2\sigma^2) + \alpha^2\sigma^2,$$
$$(1 - \beta_1\sigma^2)(1 - \beta_2\sigma^2) + \alpha^2\sigma^2 < 1, \tag{3.4}$$
$$\text{GS} : |(\beta_1 + \beta_2 + \alpha^2)\sigma^2 - 2| < 1 + (1 - \beta_1\sigma^2)(1 - \beta_2\sigma^2),$$
$$(1 - \beta_1\sigma^2)(1 - \beta_2\sigma^2) < 1. \tag{3.5}$$

*If $\beta_1 + \beta_2 + \alpha^2 < 2/\sigma_1^2$, the convergence region of GS updates **strictly** include that of Jacobi updates.*

Both characteristic polynomials can be written as a quadratic polynomial $\lambda^2 + a\lambda + b$, where:

$$\text{J: } a = (\beta_1 + \beta_2)\sigma^2 - 2, \ b = (1 - \beta_1\sigma^2)(1 - \beta_2\sigma^2) + \alpha^2\sigma^2, \tag{C.19}$$
$$\text{GS: } a = (\beta_1 + \beta_2 + \alpha^2)\sigma^2 - 2, \ b = (1 - \beta_1\sigma^2)(1 - \beta_2\sigma^2). \tag{C.20}$$

Compared to Jacobi EG, the only difference between Gauss–Seidel and Jacobi updates is that the $\alpha^2\sigma^2$ in $b$ is now in $a$, which agrees with Theorem 2.3. Using Corollary 2.1, we can derive the Schur conditions equation 3.4 and equation 3.5.

More can be said if $\beta_1 + \beta_2$ is small. For instance, if $\beta_1 + \beta_2 + \alpha^2 < 2/\sigma_1^2$, then equation 3.4 implies equation 3.5. In this case, the first conditions of equation 3.4 and equation 3.5 are equivalent, while the second condition of equation 3.4 strictly implies that of equation 3.5. Hence, the Schur region of Gauss–Seidel updates includes that of Jacobi updates. The same holds true if $\beta_1 + \beta_2 < \frac{4}{3\sigma_1^2}$.

More precisely, to show that the GS convergence region strictly contains that of the Jacobi convergence region, simply take $\beta_1 = \beta_2 = \beta$. The Schur condition for Jacobi EG and Gauss–Seidel EG are separately:

$$\text{J: } \alpha^2\sigma^2 + (\beta\sigma^2 - 1)^2 < 1, \tag{C.21}$$
$$\text{GS: } 0 < \beta\sigma^2 < 2 \text{ and } |\alpha\sigma| < 2 - \beta\sigma^2. \tag{C.22}$$

It can be shown that if $\beta = \alpha^2/3$ and $\alpha \to 0$, equation C.21 is always violated whereas equation C.22 is always satisfied.

Conversely, we give an example when Jacobi EG converges while GS EG does not. Let $\beta_1\sigma^2 = \beta_2\sigma^2 \equiv \frac{3}{2}$, then Jacobi EG converges iff $\alpha^2\sigma^2 < \frac{3}{4}$ while GS EG converges iff $\alpha^2\sigma^2 < \frac{1}{4}$.

## C.4 Proof of Theorem 3.3: Schur conditions of OGD

In this subsection, we fill in the details of the proof of Theorem 3.3, by first deriving the Schur conditions of OGD, and then studying the relation between Jacobi OGD and GS OGD.

**Theorem 3.3** (OGD). *For generalized OGD with $\alpha_1 = \alpha_2 = \alpha$, Jacobi and Gauss–Seidel updates achieve linear convergence iff for any singular value $\sigma$ of $E$, we have:*

$$\text{J} : \begin{cases} |\beta_1\beta_2\sigma^2| < 1, \ (\alpha - \beta_1)(\alpha - \beta_2) > 0, \ 4 + (\alpha + \beta_1)(\alpha + \beta_2)\sigma^2 > 0, \\ \alpha^2\left(\beta_1^2\sigma^2 + 1\right)\left(\beta_2^2\sigma^2 + 1\right) < (\beta_1\beta_2\sigma^2 + 1)(2\alpha(\beta_1 + \beta_2) + \beta_1\beta_2(\beta_1\beta_2\sigma^2 - 3)); \end{cases} \tag{3.8}$$

$$\text{GS} : \begin{cases} (\alpha - \beta_1)(\alpha - \beta_2) > 0, \ (\alpha + \beta_1)(\alpha + \beta_2)\sigma^2 < 4, \\ (\alpha\beta_1\sigma^2 + 1)(\alpha\beta_2\sigma^2 + 1) > (1 + \beta_1\beta_2\sigma^2)^2. \end{cases} \tag{3.9}$$

*The convergence region of GS updates **strictly** include that of Jacobi updates.*

The Jacobi characteristic polynomial is now quartic in the form $\lambda^4 + a\lambda^3 + b\lambda^2 + c\lambda + d$, with

$$a = -2, \ b = \alpha^2\sigma^2 + 1, \ c = -\alpha(\beta_1 + \beta_2)\sigma^2, \ d = \beta_1\beta_2\sigma^2. \tag{C.23}$$

Comparably, the GS polynomial equation 3.7 can be reduced to a cubic one $\lambda^3 + a\lambda^2 + b\lambda + c$ with

$$a = -2 + \alpha^2\sigma^2, \ b = -\alpha(\beta_1 + \beta_2)\sigma^2 + 1, \ c = \beta_1\beta_2\sigma^2. \tag{C.24}$$

First we derive the Schur conditions equation 3.8 and equation 3.9. Note that other than Corollary 2.1, an equivalent Schur condition can be read from Cheng & Chiou (2007, Theorem 1) as:

**Theorem C.1** ([Cheng & Chiou](2007)). *A real quartic polynomial $\lambda^4 + a\lambda^3 + b\lambda^2 + c\lambda + d$ is Schur stable iff:*

$$|d| < 1, \ |a| < d + 3, \ |a + c| < b + d + 1,$$
$$(1 - d)^2 b + c^2 - a(1 + d)c - (1 + d)(1 - d)^2 + a^2 d < 0. \tag{C.25}$$

With equation C.23 and Theorem C.1, it is straightforward to derive equation 3.8. With equation C.24 and Corollary 2.1, we can derive equation 3.9 without much effort.

Now, let us study the relation between the convergence region of Jacobi OGD and GS OGD, as given in equation 3.8 and equation 3.9. Namely, we want to prove the last sentence of Theorem 3.3. The outline of our proof is as follows. We first show that each region of $(\alpha, \beta_1, \beta_2)$ described in equation 3.8 (the Jacobi region) is contained in the region described in equation 3.9 (the GS region). Since we are only studying one singular value, we slightly abuse the notations and rewrite $\beta_i \sigma$ as $\beta_i$ $(i = 1, 2)$ and $\alpha\sigma$ as $\alpha$. From equation 3.6 and equation 3.7, $\beta_1$ and $\beta_2$ can switch. WLOG, we assume $\beta_1 \geq \beta_2$. There are four cases to consider:

- $\beta_1 \geq \beta_2 > 0$. The third Jacobi condition in equation 3.8 now is redundant, and we have $\alpha > \beta_1$ or $\alpha < \beta_2$ for both methods. Solving the quadratic feasibility condition for $\alpha$ gives:

$$0 < \beta_2 < 1, \ \beta_2 \leq \beta_1 < \frac{\beta_2 + \sqrt{4 + 5\beta_2^2}}{2(1 + \beta_2^2)}, \ \beta_1 < \alpha < \frac{u + \sqrt{u^2 + tv}}{t}, \tag{C.26}$$

where $u = (\beta_1\beta_2 + 1)(\beta_1 + \beta_2)$, $v = \beta_1\beta_2(\beta_1\beta_2 + 1)(\beta_1\beta_2 - 3)$, $t = (\beta_1^2 + 1)(\beta_2^2 + 1)$. On the other hand, assume $\alpha > \beta_1$, the first and third GS conditions are automatic. Solving the second gives:

$$0 < \beta_2 < 1, \ \beta_2 \leq \beta_1 < \frac{-\beta_2 + \sqrt{8 + \beta_2^2}}{2}, \ \beta_1 < \alpha < -\frac{1}{2}(\beta_1 + \beta_2) + \frac{1}{2}\sqrt{(\beta_1 - \beta_2)^2 + 16}. \tag{C.27}$$

Define $f(\beta_2) := -\beta_2 + \sqrt{8 + \beta_2^2}/2$ and $g(\beta_2) := (\beta_2 + \sqrt{4 + 5\beta_2^2})/(2(1 + \beta_2^2))$, and one can show that

$$f(\beta_2) \geq g(\beta_2). \tag{C.28}$$

Furthermore, it can also be shown that given $0 < \beta_2 < 1$ and $\beta_2 \leq \beta_1 < g(\beta_2)$, we have

$$(u + \sqrt{u^2 + 4v})/t < -(\beta_1 + \beta_2)/2 + (1/2)\sqrt{(\beta_1 - \beta_2)^2 + 16}. \tag{C.29}$$

- $\beta_1 \geq \beta_2 = 0$. The Schur condition for Jacobi and Gauss–Seidel updates reduces to:

$$\text{Jacobi: } 0 < \beta_1 < 1, \ \beta_1 < \alpha < \frac{2\beta_1}{1 + \beta_1^2}, \tag{C.30}$$

$$\text{GS: } 0 < \beta_1 < \sqrt{2}, \ \beta_1 < \alpha < \frac{-\beta_1 + \sqrt{16 + \beta_1^2}}{2}. \tag{C.31}$$

One can show that given $\beta_1 \in (0, 1)$, we have $2\beta_1/(1 + \beta_1^2) < (-\beta_1 + \sqrt{16 + \beta_1^2})/2$.

- $\beta_1 \geq 0 > \beta_2$. Reducing the first, second and fourth conditions of equation 3.8 yields:

$$\beta_2 < 0, \ 0 < \beta_1 < \frac{\beta_2 + \sqrt{4 + 5\beta_2^2}}{2(1 + \beta_2^2)}, \ \beta_1 < \alpha < \frac{u + \sqrt{u^2 + tv}}{t}. \tag{C.32}$$

This region contains the Jacobi region. It can be similarly proved that even within this larger region, GS Schur condition equation 3.9 is always satisfied.

- $\beta_2 \leq \beta_1 < 0$. We have $u < 0$, $tv < 0$ and thus $\alpha < (u + \sqrt{u^2 + tv})/t < 0$. This contradicts our assumption that $\alpha > 0$.

Combining the four cases above, we know that the Jacobi region is contained in the GS region.

To show the strict inclusion, take $\beta_1 = \beta_2 = \alpha/5$ and $\alpha \to 0$. One can show that as long as $\alpha$ is small enough, all the Jacobi regions do not contain this point, each of which is described with a

singular value in equation 3.8. However, all the GS regions described in equation 3.9 contain this point.

The proof above is still missing some details. We provide the proofs of equation C.26, equation C.28, equation C.29 and equation C.32 in the sub-sub-sections below, with the help of Mathematica, although one can also verify these claims manually. Moreover, a one line proof of the inclusion can be given with Mathematica code, as shown in Section C.4.5.

### C.4.1 PROOF OF EQUATION C.26

The fourth condition of equation 3.8 can be rewritten as:

$$\alpha^2 t - 2u\alpha - v < 0, \tag{C.33}$$

where $u = (\beta_1\beta_2 + 1)(\beta_1 + \beta_2)$, $v = \beta_1\beta_2(\beta_1\beta_2 + 1)(\beta_1\beta_2 - 3)$, $t = (\beta_1^2 + 1)(\beta_2^2 + 1)$. The discriminant is $4(u^2 + tv) = (1 - \beta_1\beta_2)^2(1 + \beta_1\beta_2)(\beta_1^2 + \beta_2^2 + \beta_1^2\beta_2^2 - \beta_1\beta_2) \geq 0$. Since if $\beta_1\beta_2 < 0$,

$$\beta_1^2 + \beta_2^2 + \beta_1^2\beta_2^2 - \beta_1\beta_2 = \beta_1^2 + \beta_2^2 + \beta_1\beta_2(\beta_1\beta_2 - 1) > 0,$$

If $\beta_1\beta_2 \geq 0$,

$$\beta_1^2 + \beta_2^2 + \beta_1^2\beta_2^2 - \beta_1\beta_2 = (\beta_1 - \beta_2)^2 + \beta_1\beta_2(1 + \beta_1\beta_2) \geq 0,$$

where we used $|\beta_1\beta_2| < 1$ in both cases. So, equation C.33 becomes:

$$\frac{u - \sqrt{u^2 + tv}}{t} < \alpha < \frac{u + \sqrt{u^2 + tv}}{t}. \tag{C.34}$$

Combining with $\alpha > \beta_1$ or $\alpha < \beta_2$ obtained from the second condition, we have:

$$\frac{u - \sqrt{u^2 + tv}}{t} < \alpha < \beta_2 \text{ or } \beta_1 < \alpha < \frac{u + \sqrt{u^2 + tv}}{t}. \tag{C.35}$$

The first case is not possible, with the following code:

```
u = (b1 b2 + 1) (b1 + b2); v = b1 b2 (b1 b2 - 3);
t = (b1^2 + 1) (b2^2  + 1);
Reduce[b2 t > u - Sqrt[u^2 + t v] && b1 >= b2 > 0
&& Abs[b1 b2] < 1],
```

and we have:

```
False.
```

Therefore, the only possible case is $\beta_1 < \alpha < (u + \sqrt{u^2 + tv})/t$. Where the feasibility region can be solved with:

```
Reduce[b1 t < u + Sqrt[u^2+t v]&&b1>=b2>0&&Abs[b1 b2] < 1].
```

What we get is:

```
0<b2<1 &&
b2<=b1<b2/(2 (1+b2^2))+1/2 Sqrt[(4+5 b2^2)/(1+b2^2)^2].
```

Therefore, we have proved equation C.26.

### C.4.2 PROOF OF EQUATION C.28

With

```
Reduce[-(b2/2) + Sqrt[8 + b2^2]/2 >=
(b2 + Sqrt[4 + 5 b2^2])/(2 (1 + b2^2)) && 0 < b2 < 1],
```

we can remove the first constraint and get:

```
0 < b2 < 1.
```

### C.4.3 PROOF OF EQUATION C.29

Given

```
   Reduce[-1/2 (b1 + b2) + 1/2 Sqrt[(b1 - b2)^2 + 16] >
   (u + Sqrt[u^2 + t v])/t &&
 0 < b2 < 1 &&
 b2 <= b1 < (b2 + Sqrt[4 + 5 b2^2])/(2 (1 + b2^2)), {b2, b1}],
```

we can remove the first constraint and get:

```
   0 < b2 < 1 &&
 b2 <= b1 < b2/(2 (1 + b2^2)) +
 1/2 Sqrt[(4 + 5 b2^2)/(1 + b2^2)^2].
```

### C.4.4 PROOF OF EQUATION C.32

The second Jacobi condition simplifies to $\alpha > \beta_1$ and the fourth simplifies to equation C.34. Combining with the first Jacobi condition:

```
   Reduce[Abs[b1 b2] < 1 &&
  a > b1 && (u - Sqrt[u^2 + t v])/t < a < (u + Sqrt[u^2 + t v])/t
  && b1 >= 0 && b2 < 0, {b2, b1, a} ] // Simplify,
```

we have:

```
   b2 < 0 && b1 > 0 &&
 b2/(1 + b2^2) + Sqrt[(4 + 5 b2^2)/(1 + b2^2)^2] > 2 b1 &&
 b1 < a < (b1 + b2 + b1^2 b2 + b1 b2^2)/((1 + b1^2) (1 + b2^2)) +
   Sqrt[((-1 + b1 b2)^2 (b1^2 + b2^2 + b1 b2 (-1 + b2^2) +
     b1^3 (b2 + b2^3)))/((1 + b1^2)^2 (1 + b2^2)^2)].
```

This can be further simplified to achieve equation C.32.

### C.4.5 ONE LINE PROOF

In fact, there is another very simple proof:

```
   Reduce[ForAll[{b1, b2, a}, (a - b1) (a - b2) > 0
   && (a + b1) (a + b2) > -4 && Abs[b1 b2] < 1 &&
   a^2 (b1^2 + 1) (b2^2 + 1) < (b1 b2 + 1) (2 a (b1 + b2) +
   b1 b2 (b1 b2 - 3)), (a - b1) (a - b2) > 0 &&
   (a + b1) (a + b2) < 4
   && (a b1 + 1) (a b2 + 1) > (1 + b1 b2)^2], {b2, b1, a}]
   True.
```

However, this proof does not tell us much information about the range of our variables.

### C.5 PROOF OF THEOREM 3.4: SCHUR CONDITIONS OF MOMENTUM

**Theorem 3.4 (momentum).** *For the generalized momentum method with $\alpha_1 = \alpha_2 = \alpha$, the Jacobi updates never converge, while the GS updates converge iff for any singular value $\sigma$ of $\mathbf{E}$, we have:*

$$|\beta_1\beta_2| < 1, |-\alpha^2\sigma^2 + \beta_1 + \beta_2 + 2| < \beta_1\beta_2 + 3, 4(\beta_1 + 1)(\beta_2 + 1) > \alpha^2\sigma^2,$$
$$\alpha^2\sigma^2\beta_1\beta_2 < (1 - \beta_1\beta_2)(2\beta_1\beta_2 - \beta_1 - \beta_2). \tag{3.12}$$

*This condition implies that at least one of $\beta_1, \beta_2$ is **negative**.*

### C.5.1 SCHUR CONDITIONS OF JACOBI AND GS UPDATES

**Jacobi condition** We first rename $\alpha\sigma$ as al and $\beta_1, \beta_2$ as b1, b2. With Theorem C.1:

```
  {Abs[d] < 1, Abs[a] < d + 3,
   a + b + c + d + 1 > 0, -a + b - c + d + 1 >
    0, (1 - d)^2 b  - (c - a d) (a - c) - (1 + d) (1 - d)^2 <
    0} /. {a -> -2 - b1 - b2, b -> al^2 + 1 + 2 (b1 + b2) + b1 b2,
   c -> -b1 - b2 - 2 b1 b2, d -> b1 b2} // FullSimplify.
```

We obtain:

```
   {Abs[b1 b2] < 1, Abs[2 + b1 + b2] < 3 + b1 b2, al^2 > 0,
 al^2 + 4 (1 + b1) (1 + b2) > 0, al^2 (-1 + b1 b2)^2 < 0}.
```

The last condition is never satisfied and thus Jacobi momentum never converges.

**Gauss–Seidel condition**   With Theorem C.1, we compute:

```
{Abs[d] < 1, Abs[a] < d + 3,
   a + b + c + d + 1 > 0, -a + b - c + d + 1 >
    0, (1 - d)^2 b  + c^2 - a (1 + d) c - (1 + d) (1 - d)^2 + a^2 d <
    0} /. {a -> al^2 - 2 - b1 - b2, b -> 1 + 2 (b1 + b2) + b1 b2,
   c -> -b1 - b2 - 2 b1 b2, d -> b1 b2} // FullSimplify.
```

The result is:

```
{Abs[b1 b2] < 1, Abs[2 - al^2 + b1 + b2] < 3 + b1 b2, al^2 > 0,
 4 (1 + b1) (1 + b2) > al^2,
 al^2 (b1 + b2 + (-2 + al^2 - b1) b1 b2 + b1 (-1 + 2 b1) b2^2) < 0},
```

which can be further simplified to equation 3.12.

### C.5.2   NEGATIVE MOMENTUM

With Theorem 3.4, we can actually show that in general at least one of $\beta_1$ and $\beta_2$ must be negative. There are three cases to consider, and in each case we simplify equation 3.12:

1. $\beta_1\beta_2 = 0$. WLOG, let $\beta_2 = 0$, and we obtain
$$-1 < \beta_1 < 0 \text{ and } \alpha^2\sigma^2 < 4(1+\beta_1). \tag{C.36}$$

2. $\beta_1\beta_2 > 0$. We have
$$-1 < \beta_1 < 0, -1 < \beta_2 < 0, \alpha^2\sigma^2 < 4(1+\beta_1)(1+\beta_2). \tag{C.37}$$

3. $\beta_1\beta_2 < 0$. WLOG, we assume $\beta_1 \geq \beta_2$. We obtain:
$$-1 < \beta_2 < 0, 0 < \beta_1 < \min\left\{-\frac{1}{3\beta_2}, \left|-\frac{\beta_2}{1+2\beta_2}\right|\right\}. \tag{C.38}$$

The constraints for $\alpha$ are $\alpha > 0$ and:
$$\max\left\{\frac{(1-\beta_1\beta_2)(2\beta_1\beta_2-\beta_1-\beta_2)}{\beta_1\beta_2}, 0\right\} < \alpha^2\sigma^2 < 4(1+\beta_1)(1+\beta_2). \tag{C.39}$$

These conditions can be further simplified by analyzing all singular values. They only depend on $\sigma_1$ and $\sigma_n$, the largest and the smallest singular values. Now, let us derive equation C.37, equation C.38 and equation C.39 more carefully. Note that we use a for $\alpha\sigma$.

### C.5.3   PROOF OF EQUATION C.37

```
   Reduce[Abs[b1 b2] < 1 && Abs[-a^2 + b1 + b2 + 2] < b1 b2 + 3 &&
 4 (b1 + 1) (b2 + 1) > a^2 &&
 a^2 b1 b2 < (1 - b1 b2) (2 b1 b2 - b1 - b2) && b1 b2 > 0 &&
 a > 0, {b2, b1, a}]

   -1 < b2 < 0 && -1 < b1 < 0 && 0 < a < Sqrt[4 + 4 b1 + 4 b2 + 4 b1 b2]
```

### C.5.4 PROOF OF EQUATIONS C.38 AND C.39

```
   Reduce[Abs[b1 b2] < 1 && Abs[-a^2 + b1 + b2 + 2] < b1 b2 + 3 &&
4 (b1 + 1) (b2 + 1) > a^2 &&
a^2 b1 b2 < (1 - b1 b2) (2 b1 b2 - b1 - b2) && b1 b2 < 0 &&
b1 >= b2 && a > 0, {b2, b1, a}]

  (-1 < b2 <= -(1/3) && ((0 < b1 <= b2/(-1 + 2 b2) &&
     0 < a < Sqrt[4 + 4 b1 + 4 b2 + 4 b1 b2]) || (b2/(-1 + 2 b2) <
      b1 < -(1/(3 b2)) &&
     Sqrt[(-b1 - b2 + 2 b1 b2 + b1^2 b2 + b1 b2^2 - 2 b1^2 b2^2)/(
      b1 b2)] < a < Sqrt[4 + 4 b1 + 4 b2 + 4 b1 b2]))) || (-(1/3) <
   b2 < 0 && ((0 < b1 <= b2/(-1 + 2 b2) &&
     0 < a < Sqrt[4 + 4 b1 + 4 b2 + 4 b1 b2]) || (b2/(-1 + 2 b2) <
      b1 < -(b2/(1 + 2 b2)) &&
     Sqrt[(-b1 - b2 + 2 b1 b2 + b1^2 b2 + b1 b2^2 - 2 b1^2 b2^2)/(
      b1 b2)] < a < Sqrt[4 + 4 b1 + 4 b2 + 4 b1 b2])))
```

Some further simplication yields equation C.38 and equation C.39.

## D PROOFS IN SECTION 4

For bilinear games and gradient-based methods, a Schur condition defines the region of convergence in the parameter space, as we have seen in Section 3. However, it is unknown which setting of parameters has the best convergence rate in a Schur stable region. We explore this problem now. Due to Theorem 3.1, we do not need to study GD. The remaining cases are EG, OGD and GS momentum (Jacobi momentum does not converge due to Theorem 3.4). Analytically (Section D.1 and D.2), we study the optimal linear rates for EG and special cases of generalized OGD (Jacobi OGD with $\beta_1 = \beta_2$ and Gauss–Seidel OGD with $\beta_2 = 0$). The special cases include the original form of OGD. We also provide details for the numerical method described at the end of Section 4.

The optimal spectral radius is obtained by solving another min-max optimization problem:

$$\min_{\boldsymbol{\theta}} \max_{\sigma \in \mathrm{Sv}(\boldsymbol{E})} r(\boldsymbol{\theta}, \sigma), \tag{D.1}$$

where $\boldsymbol{\theta}$ denotes the collection of all hyper-parameters, and $r(\boldsymbol{\theta}, \sigma)$ is defined as the spectral radius function that relies on the choice of parameters and the singular value $\sigma$. We also use $\mathrm{Sv}(\boldsymbol{E})$ to denote the set of singular values of $\boldsymbol{E}$.

In general, the function $r(\boldsymbol{\theta}, \sigma)$ is non-convex and thus difficult to analyze. However, in the special case of quadratic characteristic polynomials, it is possible to solve equation D.1. This is how we will analyze EG and special cases of OGD, as $r(\boldsymbol{\theta}, \sigma)$ can be expressed using root functions of quadratic polynomials. For cubic and quartic polynomials, it is in principle also doable as we have analytic formulas for the roots. However, these formulas are extremely complicated and difficult to optimize and we leave it for future work. For EG and OGD, we will show that the optimal linear rates depend only on the conditional number $\kappa := \sigma_1/\sigma_n$.

For simplicity, we always fix $\alpha_1 = \alpha_2 = \alpha > 0$ using the scaling symmetry studied in Section 3.

### D.1 PROOF OF THEOREM 4.1: OPTIMAL CONVERGENCE RATE OF EG

**Theorem 4.1** (**EG optimal**). *Both Jacobi and GS EG achieve the optimal exponent of linear convergence $r_* = (\kappa^2 - 1)/(\kappa^2 + 1)$ at $\alpha \to 0$ and $\beta_1 = \beta_2 = 2/(\sigma_1^2 + \sigma_n^2)$. As $\kappa \to \infty$, $r_* \to 1 - 2/\kappa^2$.*

### D.1.1 JACOBI EG

For Jacobi updates, if $\beta_1 = \beta_2 = \beta$, by solving the roots of equation 3.2, the min-max problem is:

$$\min_{\alpha, \beta} \max_{\sigma \in \mathrm{Sv}(\boldsymbol{E})} \sqrt{\alpha^2 \sigma^2 + (1 - \beta \sigma^2)^2}. \tag{D.2}$$

If $\sigma_1 = \sigma_n = \sigma$, we can simply take $\alpha \to 0$ and $\beta = 1/\sigma^2$ to obtain a super-linear convergence rate. Otherwise, let us assume $\sigma_1 > \sigma_n$. We obtain a lower bound by taking $\alpha \to 0$ and equation D.2 reduces to:

$$\min_\beta \max_{\sigma \in \mathrm{Sv}(\boldsymbol{E})} |1 - \beta\sigma^2|. \tag{D.3}$$

The optimal solution is given at $1 - \beta\sigma_n^2 = \beta\sigma_1^2 - 1$, yielding $\beta = 2/(\sigma_1^2 + \sigma_n^2)$. The optimal radius is thus $(\sigma_1^2 - \sigma_n^2)/(\sigma_1^2 + \sigma_n^2)$ since the lower bound equation D.3 can be achieved by taking $\alpha \to 0$.

From general $\beta_1$, $\beta_2$, it can be verified that the optimal radius is achieved at $\beta_1 = \beta_2$ and the problem reduces to the previous case. The optimization problem is:

$$\min_{\alpha,\beta_1,\beta_2} \max_{\sigma \in \mathrm{Sv}(\boldsymbol{E})} r(\alpha, \beta_1, \beta_2, \sigma), \tag{D.4}$$

where

$$r(\alpha, \beta_1, \beta_2, \sigma) = \begin{cases} \sqrt{(1 - \beta_1\sigma^2)(1 - \beta_2\sigma^2) + \alpha^2\sigma^2} & 4\alpha^2 > (\beta_1 - \beta_2)^2\sigma^2, \\ |1 - \frac{1}{2}(\beta_1 + \beta_2)\sigma^2| + \frac{1}{2}\sqrt{(\beta_1 - \beta_2)^2\sigma^4 - 4\alpha^2\sigma^2} & 4\alpha^2 \le (\beta_1 - \beta_2)^2\sigma^2. \end{cases}$$

In the first case, a lower bound is obtained at $\alpha^2 = (\beta_1 - \beta_2)^2\sigma^2/4$ and thus the objective only depends on $\beta_1 + \beta_2$. In the second case, the lower bound is obtained at $\alpha \to 0$ and $\beta_1 \to \beta_2$. Therefore, the function is optimized at $\beta_1 = \beta_2$ and $\alpha \to 0$.

Our analysis above does not mean that $\alpha \to 0$ and $\beta_1 = \beta_2 = 2/(\sigma_1^2 + \sigma_n^2)$ is the only optimal choice. For example, when $\sigma_1 = \sigma_n = 1$, we can take $\beta_1 = 1 + \alpha$ and $\beta_2 = 1 - \alpha$ to obtain a super-linear convergence rate.

### D.1.2   GAUSS–SEIDEL EG

For Gauss–Seidel updates and $\beta_1 = \beta_2 = \beta$, we do the following optimization:

$$\min_{\alpha,\beta} \max_{\sigma \in \mathrm{Sv}(\boldsymbol{E})} r(\alpha, \beta, \sigma), \tag{D.5}$$

where by solving equation 3.3:

$$r(\alpha, \beta, \sigma) = \begin{cases} 1 - \beta\sigma^2 & \alpha^2\sigma^2 < 4(1 - \beta\sigma^2), \\ \frac{\alpha^2}{2}\sigma^2 - (1 - \beta\sigma^2) + \sqrt{\alpha^2\sigma^2(\alpha^2\sigma^2 - 4(1 - \beta\sigma^2))}/2 & \alpha^2\sigma^2 \ge 4(1 - \beta\sigma^2). \end{cases}$$

$r(\sigma, \beta, \sigma^2)$ is quasi-convex in $\sigma^2$, so we just need to minimize over $\alpha, \beta$ at both end points. Hence, equation D.5 reduces to:

$$\min_{\alpha,\beta} \max\{r(\alpha, \beta, \sigma_1), r(\alpha, \beta, \sigma_n)\}.$$

By arguing over three cases: $\alpha^2 + 4\beta < 4/\sigma_1^2$, $\alpha^2 + 4\beta > 4/\sigma_n^2$ and $4/\sigma_1^2 \le \alpha^2 + 4\beta \le 4/\sigma_n^2$, we find that the minimum $(\kappa^2 - 1)/(\kappa^2 + 1)$ can be achieved at $\alpha \to 0$ and $\beta = 2/(\sigma_1^2 + \sigma_n^2)$, the same as Jacobi EG. This is because $\alpha \to 0$ decouples $x$ and $y$ and it does not matter whether the update is Jacobi or GS.

For general $\beta_1$, $\beta_2$, it can be verified that the optimal radius is achieved at $\beta_1 = \beta_2$. We do the following transformation: $\beta_i \to \xi_i - \alpha^2/2$, so that the characteristic polynomial becomes:

$$(\lambda - 1)^2 + (\xi_1 + \xi_2)\sigma^2(\lambda - 1) + \alpha^2\sigma^2 + (\xi_1 - \alpha^2/2)(\xi_2 - \alpha^2/2)\sigma^4 = 0. \tag{D.6}$$

Denote $\xi_1 + \xi_2 = \phi$, and $(\xi_1 - \alpha^2/2)(\xi_2 - \alpha^2/2) = \nu$, we have:

$$\lambda^2 - (2 - \sigma^2\phi)\lambda + 1 - \sigma^2\phi + \sigma^4\nu + \sigma^2\alpha^2 = 0. \tag{D.7}$$

The discriminant is $\Delta := \sigma^2(\sigma^2(\phi^2 - 4\nu) - 4\alpha^2)$. We discuss two cases:

1. $\phi^2 - 4\nu < 0$. We are minimizing:

$$\min_{\alpha,u,v} \sqrt{1 + (\alpha^2 - \phi)\sigma_1^2 + \sigma_1^4\nu} \vee \sqrt{1 + (\alpha^2 - \phi)\sigma_n^2 + \sigma_n^4\nu},$$

   with $a \vee b := \max\{a, b\}$ a shorthand. A minimizer is at $\alpha \to 0$ and $\nu \to \phi^2/4$ (since $\phi^2 < 4\nu$), where $\beta_1 = \beta_2 = 2/(\sigma_1^2 + \sigma_n^2)$ and $\alpha \to 0$.

2. $\phi^2 - 4\nu \geq 0$. A lower bound is:
$$\min_u |1 - \phi\sigma_1^2/2| \vee |1 - \phi\sigma_n^2/2|,$$
which is obtained iff $4\alpha^2 \sim (\phi^2 - 4\nu)t$ for all $\sigma^2$. This is only possible if $\alpha \to 0$ and $\phi^2 \to 4\nu$, which yields $\beta_1 = \beta_2 = 2/(\sigma_1^2 + \sigma_n^2)$.

From what has been discussed, the optimal radius is $(\kappa^2 - 1)/(\kappa^2 + 1)$ which can be achieved at $\beta_1 = \beta_2 = 2/(\sigma_1^2 + \sigma_n^2)$ and $\alpha \to 0$. Again, this might not be the only choice. For instance, take $\sigma_1 = \sigma_n^2 = 1$, from equation 3.3, a super-linear convergence rate can be achieved at $\beta_1 = 1$ and $\beta_2 = 1 - \alpha^2$.

## D.2 Proof of Theorem 4.2: Optimal convergence rate of OGD

**Theorem 4.2** (**OGD optimal**). *For Jacobi OGD with $\beta_1 = \beta_2 = \beta$, to achieve the optimal linear rate, we must have $\alpha \leq 2\beta$. For the original OGD with $\alpha = 2\beta$, the optimal linear rate $r_*$ satisfies*

$$r_*^2 = \frac{1}{2} + \frac{1}{4\sqrt{2}\sigma_1^2}\sqrt{(\sigma_1^2 - \sigma_n^2)(5\sigma_1^2 - \sigma_n^2 + \sqrt{(\sigma_1^2 - \sigma_n^2)(9\sigma_1^2 - \sigma_n^2)})}, \tag{D.8}$$

*at*

$$\beta_* = \frac{1}{4\sqrt{2}}\sqrt{\frac{3\sigma_1^4 - (\sigma_1^2 - \sigma_n^2)^{3/2}\sqrt{9\sigma_1^2 - \sigma_n^2} + 6\sigma_1^2\sigma_n^2 - \sigma_n^4}{\sigma_1^4\sigma_n^2}}. \tag{D.9}$$

*If $\kappa \to \infty$, $r_* \sim 1 - 1/(6\kappa^2)$. For Gauss–Seidel OGD with $\beta_2 = 0$, the optimal linear rate is $r_* = \sqrt{(\kappa^2 - 1)/(\kappa^2 + 1)}$, at $\alpha = \sqrt{2}/\sigma_1$ and $\beta_1 = \sqrt{2}\sigma_1/(\sigma_1^2 + \sigma_n^2)$. If $\kappa \to \infty$, $r_* \sim 1 - 1/\kappa^2$.*

For OGD, the characteristic polynomials equation 3.6 and equation 3.7 are quartic and cubic separately, and thus optimizing the spectral radii for generalized OGD is difficult. However, we can study two special cases: for Jacobi OGD, we take $\beta_1 = \beta_2$; for Gauss–Seidel OGD, we take $\beta_2 = 0$. In both cases, the spectral radius functions can be obtained by solving quadratic polynomials.

### D.2.1 Jacobi OGD

We assume $\beta_1 = \beta_2 = \beta$ in this subsection. The characteristic polynomial for Jacobi OGD equation 3.6 can be written as:
$$\lambda^2(\lambda - 1)^2 + (\lambda\alpha - \beta)^2\sigma^2 = 0. \tag{D.10}$$
Factorizing it gives two equations which are conjugate to each other:
$$\lambda(\lambda - 1) \pm i(\lambda\alpha - \beta)\sigma = 0. \tag{D.11}$$
The roots of one equation are the conjugates of the other equation. WLOG, we solve $\lambda(\lambda - 1) + i(\lambda\alpha - \beta)\sigma = 0$ which gives $(1/2)(u \pm v)$, where
$$u = 1 - i\alpha\sigma, \ v = \sqrt{1 - \alpha^2\sigma^2 - 2i(\alpha - 2\beta)\sigma}. \tag{D.12}$$

Denote $\Delta_1 = 1 - \alpha^2\sigma^2$ and $\Delta_2 = 2(\alpha - 2\beta)\sigma$. If $\alpha \geq 2\beta$, $v$ can be expressed as:
$$v = \frac{1}{\sqrt{2}}\left(\sqrt{\sqrt{\Delta_1^2 + \Delta_2^2} + \Delta_1} - i\sqrt{\sqrt{\Delta_1^2 + \Delta_2^2} - \Delta_1}\right) =: \frac{1}{\sqrt{2}}(a - ib), \tag{D.13}$$
therefore, the spectral radius $r(\alpha, \beta, \sigma)$ satisfies:
$$r(\alpha, \beta, \sigma)^2 = \frac{1}{4}\left((1 + a/\sqrt{2})^2 + (\alpha\sigma + b/\sqrt{2})^2\right) = \frac{1}{4}(1 + \alpha^2\sigma^2 + \sqrt{\Delta_1^2 + \Delta_2^2} + \sqrt{2}(b\sigma\alpha + a)), \tag{D.14}$$

and the minimum is achieved at $\alpha = 2\beta$. From now on, we assume $\alpha \leq 2\beta$, and thus $v = a + ib$. We write:

$$\begin{aligned}
r(\alpha, \beta, \sigma)^2 &= \frac{1}{4}\max\{\left((1 + a/\sqrt{2})^2 + (\alpha\sigma - b/\sqrt{2})^2\right), \left((1 - a/\sqrt{2})^2 + (\alpha\sigma + b/\sqrt{2})^2\right)\}, \\
&= \frac{1}{4}(1 + \alpha^2\sigma^2 + \sqrt{\Delta_1^2 + \Delta_2^2} + \sqrt{2}|b\sigma\alpha - a|). \\
&= \begin{cases} \frac{1}{4}(1 + \alpha^2\sigma^2 + \sqrt{\Delta_1^2 + \Delta_2^2} - \sqrt{2}(b\sigma\alpha - a)) & 0 < \alpha\sigma \leq 1, \\ \frac{1}{4}(1 + \alpha^2\sigma^2 + \sqrt{\Delta_1^2 + \Delta_2^2} + \sqrt{2}(b\sigma\alpha - a)) & \alpha\sigma > 1. \end{cases}
\end{aligned} \tag{D.15}$$

This is a non-convex and non-differentiable function, which is extremely difficult to optimize.

At $\alpha = 2\beta$, in this case, $a = \sqrt{1 - 4\beta^2\sigma^2}\text{sign}(1 - 4\beta^2\sigma^2)$ and $b = \sqrt{4\beta^2\sigma^2 - 1}\text{sign}(4\beta^2\sigma^2 - 1)$. The sign function $\text{sign}(x)$ is defined to be 1 if $x > 0$ and 0 otherwise. The function we are optimizing is a quasi-convex function:

$$r(\beta, \sigma)^2 = \begin{cases} \frac{1}{2}(1 + \sqrt{1 - 4\beta^2\sigma^2}) & 4\beta^2\sigma^2 \le 1, \\ 2\beta^2\sigma^2 + \beta\sigma\sqrt{4\beta^2\sigma^2 - 1} & 4\beta^2\sigma^2 > 1. \end{cases} \tag{D.16}$$

We are maximizing over $\sigma$ and minimizing over $\beta$. There are three cases:

- $4\beta^2\sigma_1^2 \le 1$. At $4\beta^2\sigma_1^2 = 1$, the optimal radius is:

$$r_*^2 = \frac{1}{2}\left(1 + \sqrt{1 - \frac{1}{\kappa^2}}\right).$$

- $4\beta^2\sigma_n^2 \ge 1$. At $4\beta^2\sigma_n^2 = 1$, the optimal radius satisfies:

$$r_*^2 = \frac{\kappa^2}{2} + \frac{\kappa}{2}\sqrt{\kappa^2 - 1}.$$

- $4\beta^2\sigma_n^2 \le 1$ and $4\beta^2\sigma_1^2 \ge 1$. The optimal $\beta$ is achieved at:

$$\frac{1}{2}\left(1 + \sqrt{1 - 4\beta^2\sigma_n^2}\right) = 2\beta^2\sigma_1^2 + \beta\sigma_1\sqrt{4\beta^2\sigma_1^2 - 1}.$$

  The solution is unique since the left is decreasing and the right is increasing. The optimal $\beta$ is:

$$\beta_* = \frac{1}{4\sqrt{2}}\sqrt{\frac{3\sigma_1^4 - (\sigma_1^2 - \sigma_n^2)^{3/2}\sqrt{9\sigma_1^2 - \sigma_n^2} + 6\sigma_1^2\sigma_n^2 - \sigma_n^4}{\sigma_1^4\sigma_n^2}}. \tag{D.17}$$

  The optimal radius satisfies:

$$r_*^2 = \frac{1}{2} + \frac{1}{4\sqrt{2}\sigma_1^2}\sqrt{(\sigma_1^2 - \sigma_n^2)(5\sigma_1^2 - \sigma_n^2 + \sqrt{(\sigma_1^2 - \sigma_n^2)(9\sigma_1^2 - \sigma_n^2)})}. \tag{D.18}$$

  This is the optimal solution among the three cases. If $\sigma_n^2/\sigma_1^2$ is small enough we have $r^2 \sim 1 - 1/(3\kappa^2)$.

### D.2.2 GAUSS–SEIDEL OGD

In this subsection, we study Gauss–Seidel OGD and fix $\beta_2 = 0$. The characteristic polynomial equation 3.7 now reduces to a quadratic polynomial:

$$\lambda^2 + (\alpha^2\sigma^2 - 2)\lambda + 1 - \alpha\beta_1\sigma^2 = 0.$$

For convenience, we reparametrize $\beta_1 \to \beta/\alpha$. So, the quadratic polynomial becomes:

$$\lambda^2 + (\alpha^2\sigma^2 - 2)\lambda + 1 - \beta\sigma^2 = 0.$$

We are doing a min-max optimization $\min_{\alpha,\beta} \max_\sigma r(\alpha, \beta, \sigma)$, where $r(\alpha, \beta, \sigma)$ is:

$$r(\alpha, \beta, \sigma) = \begin{cases} \sqrt{1 - \beta\sigma^2} & \alpha^4\sigma^2 < 4(\alpha^2 - \beta) \\ \frac{1}{2}|\alpha^2\sigma^2 - 2| + \frac{1}{2}\sqrt{\alpha^4\sigma^4 - 4(\alpha^2 - \beta)\sigma^2} & \alpha^4\sigma^2 \ge 4(\alpha^2 - \beta). \end{cases} \tag{D.19}$$

There are three cases to consider:

- $\alpha^4\sigma_1^2 \le 4(\alpha^2 - \beta)$. We are minimizing $1 - \beta\sigma_n^2$ over $\alpha$ and $\beta$. Optimizing over $\beta_1$ gives $\beta = \alpha^2 - \alpha^4\sigma_1^2/4$. Then we minimize over $\alpha$ and obtain $\alpha^2 = 2/\sigma_1^2$. The optimal $\beta = 1/\sigma_1^2$ and the optimal radius is $\sqrt{1 - 1/\kappa^2}$.

- $\alpha^4 \sigma_n^2 > 4(\alpha^2 - \beta)$. Fixing $\alpha$, the optimal $\beta = \alpha^2 - \alpha^4 \sigma_n^2/4$, and we are solving

$$\min_{\alpha} \max \left\{ \frac{1}{2}|\alpha^2 \sigma_1^2 - 2| + \frac{1}{2}\alpha^2 \sqrt{\sigma_1^2(\sigma_1^2 - \sigma_n^2)}, \frac{1}{2}|\alpha^2 \sigma_n^2 - 2| \right\}.$$

  We need to discuss three cases: $\alpha^2 \sigma_n^2 > 2$, $\alpha^2 \sigma_1^2 < 2$ and $2/\sigma_1^2 < \alpha^2 < 2/\sigma_n^2$. In the first case, the optimal radius is

$$\kappa^2 - 1 + \kappa\sqrt{(\kappa^2 - 1)}.$$

  In the second case, $\alpha^2 \to 2/\sigma_1^2$ and the optimal radius is $\sqrt{1 - 1/\kappa^2}$. In the third case, the optimal radius is also $\sqrt{1 - 1/\kappa^2}$ minimized at $\alpha^2 \to 2/\sigma_1^2$.

- $\alpha^4 \sigma_1^2 > 4(\alpha^2 - \beta)$ and $\alpha^4 \sigma_n^2 < 4(\alpha^2 - \beta)$. In this case, we have $\alpha^2 \sigma_1^2 < 4$. Otherwise, $r(\alpha, \beta, \sigma_1) > 1$. We are minimizing over:

$$\max\{\sqrt{1 - \beta\sigma_n^2}, \frac{1}{2}|\alpha^2 \sigma_1^2 - 2| + \frac{1}{2}\sqrt{\alpha^4 \sigma_1^4 - 4\alpha^2 \sigma_1^2 + 4\beta\sigma_1^2}\}.$$

  The minimum over $\alpha$ is achieved at $\alpha^2 \sigma_1^2 = 2$, and $\beta = 2/(\sigma_1^2 + \sigma_n^2)$, this gives $\alpha = \sqrt{2}/\sigma_1$ and $\beta_1 = \sqrt{2}\sigma_1/(\sigma_1^2 + \sigma_n^2)$. The optimal radius is $r_* = \sqrt{(\kappa^2 - 1)/(\kappa^2 + 1)}$.

Out of the three cases, the optimal radius is obtained in the third case, where $r \sim 1 - 1/\kappa^2$. This is better than Jacobi OGD, but still worse than the optimal EG.

## D.3 NUMERICAL METHOD

We first prove Lemma 4.1:

**Lemma 4.1.** *A polynomial $p(\lambda)$ is $r$-Schur stable iff $p(r\lambda)$ is Schur stable.*

*Proof.* Denote $p(\lambda) = \prod_{i=1}^{n}(\lambda - \lambda_i)$. We have $p(r\lambda) \propto \prod_{i=1}^{n}(\lambda - \lambda_i/r)$, and:

$$\forall i \in [n], |\lambda_i| < r \iff \forall i \in [n], |\lambda_i/r| < 1. \tag{D.20}$$

$\square$

With Lemma 4.1 and Corollary 2.1, we have the following corollary:

**Corollary D.1.** *A real quadratic polynomial $\lambda^2 + a\lambda + b$ is $r$-Schur stable iff $b < r^2$, $|a| < r + b/r$; A real cubic polynomial $\lambda^3 + a\lambda^2 + b\lambda + c$ is $r$-Schur stable iff $|c| < r^3$, $|ar^2 + c| < r^3 + br$, $br^4 - acr^2 < r^6 - c^2$; A real quartic polynomial $\lambda^4 + a\lambda^3 + b\lambda^2 + c\lambda + d$ is $r$-Schur stable iff $|cr^5 - adr^3| < r^8 - d^2$, $|ar^2 + c| < br + d/r + r^3$, and*

$$b < r^2 + dr^{-2} + r^2 \frac{(cr^2 - ad)(ar^2 - c)}{(d - r^4)^2}.$$

*Proof.* In Corollary 2.1, rescale the coefficients according to Lemma 4.1. $\square$

We can use the corollaries above to find the regions where $r$-Schur stability is possible, i.e., a linear rate of exponent $r$. A simple algorithm might be to start from $r_0 = 1$, find the region $S_0$. Then recursively take $r_{t+1} = sr_t$ and find the Schur stable region $S_{t+1}$ inside $S_t$. If the region is empty then stop the search and return $S_t$. $s$ can be taken to be, say, 0.99. Formally, this algorithm can be described as follows in Algorithm 1:

$r_0 = 1$, $t = 0$, $s = 0.99$;
Find the $r_0$-Schur region $S_0$;
**while** $S_t$ *is not empty* **do**
    $r_{t+1} = sr_t$;
    Find the $r_{t+1}$-Schur region $S_{t+1}$;
    $t = t + 1$;
**end**

**Algorithm 1:** Numerical method for finding the optimal convergence rate

In this algorithm, Corollary D.1 can be applied to obtain any $r$-Schur region.

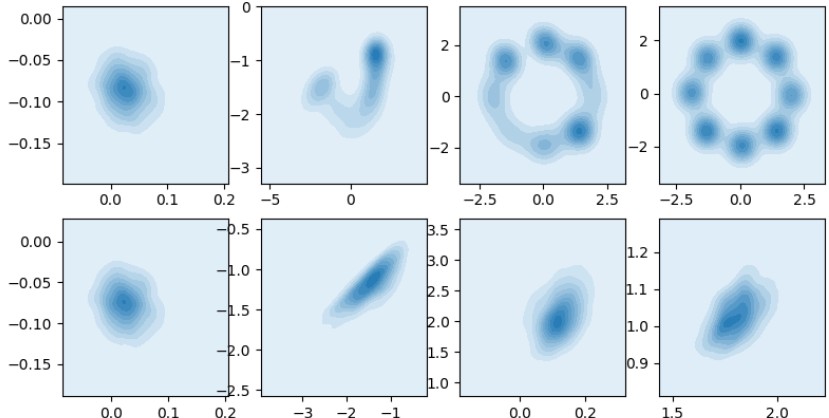

Figure 6: Test samples generated from the generator network trained with stochastic Adam. **Top row:** Jacobi updates; **Bottom row:** Gauss–Seidel updates. **Columns** (left to right): epoch 0, 5, 10, 20.

## E   Supplementary material for Sections 5 and 6

We provide supplementary material for Sections 5 and 6. We first prove that when learning the mean of a Gaussian, WGAN is locally a bilinear game in Appendix E.1. For mixtures of Gaussians, we provide supplementary experiments about Adam in Appendix E.2. This result implies that in some cases, Jacobi updates are better than GS updates. We further verify this claim in Appendix E.3 by showing an example of OGD on bilinear games. Optimizing the spectral radius given a certain singular value is possible numerically, as in Appendix E.4.

### E.1   Wasserstein GAN

Inspired by Daskalakis et al. (2018), we consider the following WGAN (Arjovsky et al., 2017):

$$f(\boldsymbol{\phi}, \boldsymbol{\theta}) = \min_{\boldsymbol{\phi}} \max_{\boldsymbol{\theta}} \mathbb{E}_{\boldsymbol{x} \sim \mathcal{N}(\boldsymbol{v}, \sigma^2 \boldsymbol{I})}[s(\boldsymbol{\theta}^\top \boldsymbol{x})] - \mathbb{E}_{\boldsymbol{z} \sim \mathcal{N}(\boldsymbol{0}, \sigma^2 \boldsymbol{I})}[s(\boldsymbol{\theta}^\top (\boldsymbol{z} + \boldsymbol{\phi}))], \qquad \text{(E.1)}$$

with $s(x) := 1/(1 + e^{-x})$ the sigmoid function. We study the local behavior near the saddle point $(\boldsymbol{v}, \boldsymbol{0})$, which depends on the Hessian:

$$\begin{bmatrix} \nabla^2_{\boldsymbol{\phi}\boldsymbol{\phi}} & \nabla^2_{\boldsymbol{\phi}\boldsymbol{\theta}} \\ \nabla^2_{\boldsymbol{\theta}\boldsymbol{\phi}} & \nabla^2_{\boldsymbol{\theta}\boldsymbol{\theta}} \end{bmatrix} = \begin{bmatrix} -\mathbb{E}_{\boldsymbol{\phi}}[s''(\boldsymbol{\theta}^\top \boldsymbol{z})\boldsymbol{\theta}\boldsymbol{\theta}^\top] & -\mathbb{E}_{\boldsymbol{\phi}}[s''(\boldsymbol{\theta}^\top \boldsymbol{z})\boldsymbol{\theta}\boldsymbol{z}^\top + s'(\boldsymbol{\theta}^\top \boldsymbol{z})\boldsymbol{I}] \\ (\nabla^2_{\boldsymbol{\phi}\boldsymbol{\theta}})^\top & \mathbb{E}_{\boldsymbol{v}}[s''(\boldsymbol{\theta}^\top \boldsymbol{x})\boldsymbol{x}\boldsymbol{x}^\top] - \mathbb{E}_{\boldsymbol{\phi}}[s''(\boldsymbol{\theta}^\top \boldsymbol{z})\boldsymbol{z}\boldsymbol{z}^\top] \end{bmatrix},$$

with $\mathbb{E}_{\boldsymbol{v}}$ a shorthand for $\mathbb{E}_{\boldsymbol{x} \sim \mathcal{N}(\boldsymbol{v}, \sigma^2 \boldsymbol{I})}$ and $\mathbb{E}_{\boldsymbol{\phi}}$ for $\mathbb{E}_{\boldsymbol{z} \sim \mathcal{N}(\boldsymbol{\phi}, \sigma^2 \boldsymbol{I})}$. At the saddle point, the Hessian is simplified as:

$$\begin{bmatrix} \nabla^2_{\boldsymbol{\phi}\boldsymbol{\phi}} & \nabla^2_{\boldsymbol{\phi}\boldsymbol{\theta}} \\ \nabla^2_{\boldsymbol{\theta}\boldsymbol{\phi}} & \nabla^2_{\boldsymbol{\theta}\boldsymbol{\theta}} \end{bmatrix} = \begin{bmatrix} \boldsymbol{0} & -s'(0)\boldsymbol{I} \\ -s'(0)\boldsymbol{I} & \boldsymbol{0} \end{bmatrix} = \begin{bmatrix} \boldsymbol{0} & -\boldsymbol{I}/4 \\ -\boldsymbol{I}/4 & \boldsymbol{0} \end{bmatrix}.$$

Therefore, this WGAN is locally a bilinear game.

### E.2   Mixtures of Gaussians with Adam

Given the same parameter settings as in Section 5, we train the vanilla GAN using Adam, with the step size $\alpha = 0.0002$, and $\beta_1 = 0.9$, $\beta_2 = 0.999$. As shown in Figure 6, Jacobi updates converge faster than the corresponding GS updates.

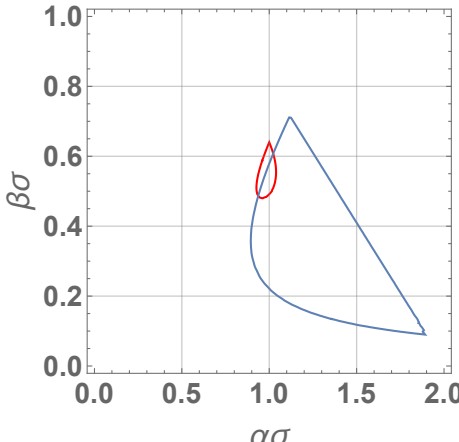

Figure 7: Contour plot of spectral radius equal to $0.8$. The red curve is for the Jacobi polynomial and the blue curve is for the GS polynomial. The GS region is larger but for some parameter settings, Jacobi OGD achieves a faster convergence rate.

### E.3 JACOBI UPDATES MAY CONVERGE FASTER THAN GS UPDATES

Take $\alpha = 0.9625$, $\beta_1 = \beta_2 = \beta = 0.5722$, and $\sigma = 1$, the Jacobi and GS OGD radii are separately $0.790283$ and $0.816572$ (by solving equation 3.6 and equation 3.7), which means that Jacobi OGD has better performance for this setting of parameters. A more intuitive picture is given as Figure 7, where we take $\beta_1 = \beta_2 = \beta$.

### E.4 SINGLE SINGULAR VALUE

We minimize $r(\boldsymbol{\theta}, \sigma)$ for a given singular value numerically. WLOG, we take $\sigma = 1$, since we can rescale parameters to obtain other values of $\sigma$. We implement grid search for all the parameters within the range $[-2, 2]$ and step size $0.05$. For the step size $\alpha$, we take it to be positive. We use $\{a, b, s\}$ as a shorthand for $\{a, a + s, a + 2s, \ldots, b\}$.

- We first numerically solve the characteristic polynomial for Jacobi OGD equation 3.6, fixing $\alpha_1 = \alpha_2 = \alpha$ with scaling symmetry. With $\alpha \in \{0, 2, 0.05\}$, $\beta_i \in \{-2, 2, 0.05\}$, the best parameter setting is $\alpha = 0.7$, $\beta_1 = 0.1$ and $\beta_2 = 0.6$. $\beta_1$ and $\beta_2$ can be switched. The optimal radius is $0.6$.

- We also numerically solve the characteristic polynomial for Gauss–Seidel OGD equation 3.7, fixing $\alpha_1 = \alpha_2 = \alpha$ with scaling symmetry. With $\alpha \in \{0, 2, 0.05\}$, $\beta_i \in \{-2, 2, 0.05\}$, the best parameter setting is $\alpha = 1.4$, $\beta_1 = 0.7$ and $\beta_2 = 0$. $\beta_1$ and $\beta_2$ can be switched. The optimal rate is $1/(5\sqrt{2})$. This rate can be further improved to be zero where $\alpha = \sqrt{2}$, $\beta_1 = 1/\sqrt{2}$ and $\beta_2 = 0$.

- Finally, we numerically solve the polynomial for Gauss–Seidel momentum equation 3.11, with the same grid. The optimal parameter choice is $\alpha = 1.8$, $\beta_1 = -0.1$ and $\beta_2 = -0.05$. $\beta_1$ and $\beta_2$ can be switched. The optimal rate is $0.5$.

## F SPLITTING METHOD

In this appendix, we interpret the gradient-based algorithms (except PP) we have studied in this paper as splitting methods (Saad, 2003), for both Jacobi and Gauss–Seidel updates. By doing this, one can understand our algorithms better in the context of numerical linear algebra and compare our results in Section 3 with the Stein–Rosenberg theorem.

### F.1 JACOBI UPDATES

From equation 2.2, finding a saddle point is equivalent to solving:

$$\boldsymbol{S}\boldsymbol{z} := \begin{bmatrix} \boldsymbol{0} & \boldsymbol{E} \\ -\boldsymbol{E}^\top & \boldsymbol{0} \end{bmatrix} \begin{bmatrix} \boldsymbol{x} \\ \boldsymbol{y} \end{bmatrix} = \begin{bmatrix} -\boldsymbol{b} \\ \boldsymbol{c} \end{bmatrix} =: \boldsymbol{d}. \tag{F.1}$$

Now, we try to understand the Jacobi algorithms using splitting method. For GD and EG, the method splits $\boldsymbol{S}$ into $\boldsymbol{M} - \boldsymbol{N}$ and solve

$$\boldsymbol{z}_{t+1} = \boldsymbol{M}^{-1}\boldsymbol{N}\boldsymbol{z}_t + \boldsymbol{M}^{-1}\boldsymbol{d}. \tag{F.2}$$

For GD, we can obtain that:

$$\boldsymbol{M} = \begin{bmatrix} \alpha_1^{-1}\boldsymbol{I} & \boldsymbol{0} \\ \boldsymbol{0} & \alpha_2^{-1}\boldsymbol{I} \end{bmatrix}, \ \boldsymbol{N} = \begin{bmatrix} \alpha_1^{-1}\boldsymbol{I} & -\boldsymbol{E} \\ \boldsymbol{E}^\top & \alpha_2^{-1}\boldsymbol{I} \end{bmatrix}. \tag{F.3}$$

For EG, we need to compute an inverse:

$$\boldsymbol{M}^{-1} = \begin{bmatrix} \alpha_1\boldsymbol{I} & -\beta_1\boldsymbol{E} \\ \beta_2\boldsymbol{E}^\top & \alpha_2\boldsymbol{I} \end{bmatrix}, \ \boldsymbol{N} = \boldsymbol{M} - \boldsymbol{S}. \tag{F.4}$$

Given $\det(\alpha_1\alpha_2\boldsymbol{I} + \beta_1\beta_2\boldsymbol{E}\boldsymbol{E}^\top) \neq 0$, the inverse always exists.

The splitting method can also work for second-step methods, such as OGD and momentum. We split $\boldsymbol{S} = \boldsymbol{M} - \boldsymbol{N} - \boldsymbol{P}$ and solve:

$$\boldsymbol{z}_{t+1} = \boldsymbol{M}^{-1}\boldsymbol{N}\boldsymbol{z}_t + \boldsymbol{M}^{-1}\boldsymbol{P}\boldsymbol{z}_{t-1} + \boldsymbol{M}^{-1}\boldsymbol{d}. \tag{F.5}$$

For OGD, we have:

$$\boldsymbol{M} = \begin{bmatrix} \frac{\boldsymbol{I}}{\alpha_1-\beta_1} & \boldsymbol{0} \\ \boldsymbol{0} & \frac{\boldsymbol{I}}{\alpha_2-\beta_2} \end{bmatrix}, \ \boldsymbol{N} = \begin{bmatrix} \frac{\boldsymbol{I}}{\alpha_1-\beta_1} & -\frac{\alpha_1\boldsymbol{E}}{\alpha_1-\beta_1} \\ \frac{\alpha_2\boldsymbol{E}^\top}{\alpha_2-\beta_2} & \frac{\boldsymbol{I}}{\alpha_2-\beta_2} \end{bmatrix}, \ \boldsymbol{P} = \begin{bmatrix} \boldsymbol{0} & \frac{\beta_1\boldsymbol{E}}{\alpha_1-\beta_1} \\ -\frac{\beta_2\boldsymbol{E}^\top}{\alpha_2-\beta_2} & \boldsymbol{0} \end{bmatrix}. \tag{F.6}$$

For the momentum method, we can write:

$$\boldsymbol{M} = \begin{bmatrix} \alpha_1^{-1}\boldsymbol{I} & \boldsymbol{0} \\ \boldsymbol{0} & \alpha_2^{-1}\boldsymbol{I} \end{bmatrix}, \ \boldsymbol{N} = \begin{bmatrix} \frac{1+\beta_1}{\alpha_1}\boldsymbol{I} & -\boldsymbol{E} \\ \boldsymbol{E}^\top & \frac{1+\beta_2}{\alpha_2}\boldsymbol{I} \end{bmatrix}, \ \boldsymbol{P} = \begin{bmatrix} -\frac{\beta_1}{\alpha_1}\boldsymbol{I} & \boldsymbol{0} \\ \boldsymbol{0} & -\frac{\beta_2}{\alpha_2}\boldsymbol{I} \end{bmatrix}. \tag{F.7}$$

### F.2 GAUSS–SEIDEL UPDATES

Now, we try to understand the GS algorithms using splitting method. For GD and EG, the method splits $\boldsymbol{S}$ into $\boldsymbol{M} - \boldsymbol{N}$ and solve

$$\boldsymbol{z}_{t+1} = \boldsymbol{M}^{-1}\boldsymbol{N}\boldsymbol{z}_t + \boldsymbol{M}^{-1}\boldsymbol{d}. \tag{F.8}$$

For GD, we can obtain that:

$$\boldsymbol{M} = \begin{bmatrix} \alpha_1^{-1}\boldsymbol{I} & \boldsymbol{0} \\ -\boldsymbol{E}^\top & \alpha_2^{-1}\boldsymbol{I} \end{bmatrix}, \ \boldsymbol{N} = \begin{bmatrix} \alpha_1^{-1}\boldsymbol{I} & -\boldsymbol{E} \\ \boldsymbol{0} & \alpha_2^{-1}\boldsymbol{I} \end{bmatrix}. \tag{F.9}$$

For EG, we need to compute an inverse:

$$\boldsymbol{M}^{-1} = \begin{bmatrix} \alpha_1\boldsymbol{I} & -\beta_1\boldsymbol{E} \\ (\beta_2+\alpha_1\alpha_2)\boldsymbol{E}^\top & \alpha_2(\boldsymbol{I}-\beta_1\boldsymbol{E}^\top\boldsymbol{E}) \end{bmatrix}, \ \boldsymbol{N} = \boldsymbol{M} - \boldsymbol{S}. \tag{F.10}$$

The splitting method can also work for second-step methods, such as OGD and momentum. We split $\boldsymbol{S} = \boldsymbol{M} - \boldsymbol{N} - \boldsymbol{P}$ and solve:

$$\boldsymbol{z}_{t+1} = \boldsymbol{M}^{-1}\boldsymbol{N}\boldsymbol{z}_t + \boldsymbol{M}^{-1}\boldsymbol{P}\boldsymbol{z}_{t-1} + \boldsymbol{M}^{-1}\boldsymbol{d}. \tag{F.11}$$

For OGD, we obtain:

$$\boldsymbol{M} = \begin{bmatrix} \frac{\boldsymbol{I}}{\alpha_1-\beta_1} & \boldsymbol{0} \\ -\frac{\alpha_2\boldsymbol{E}^\top}{\alpha_2-\beta_2} & \frac{\boldsymbol{I}}{\alpha_2-\beta_2} \end{bmatrix}, \ \boldsymbol{N} = \begin{bmatrix} \frac{\boldsymbol{I}}{\alpha_1-\beta_1} & -\frac{\alpha_1\boldsymbol{E}}{\alpha_1-\beta_1} \\ -\frac{\beta_2\boldsymbol{E}^\top}{\alpha_2-\beta_2} & \frac{\boldsymbol{I}}{\alpha_2-\beta_2} \end{bmatrix}, \ \boldsymbol{P} = \begin{bmatrix} \boldsymbol{0} & \frac{\beta_1\boldsymbol{E}}{\alpha_1-\beta_1} \\ \boldsymbol{0} & \boldsymbol{0} \end{bmatrix}. \tag{F.12}$$

For the momentum method, we can write:

$$\boldsymbol{M} = \begin{bmatrix} \alpha_1^{-1}\boldsymbol{I} & \boldsymbol{0} \\ -\boldsymbol{E}^\top & \alpha_2^{-1}\boldsymbol{I} \end{bmatrix}, \ \boldsymbol{N} = \begin{bmatrix} \frac{1+\beta_1}{\alpha_1}\boldsymbol{I} & -\boldsymbol{E} \\ \boldsymbol{0} & \frac{1+\beta_2}{\alpha_2}\boldsymbol{I} \end{bmatrix}, \ \boldsymbol{P} = \begin{bmatrix} -\frac{\beta_1}{\alpha_1}\boldsymbol{I} & \boldsymbol{0} \\ \boldsymbol{0} & -\frac{\beta_2}{\alpha_2}\boldsymbol{I} \end{bmatrix}. \tag{F.13}$$

# G    SINGULAR BILINEAR GAMES

In this paper we considered the bilinear game when $E$ is a non-singular square matrix for simplicity. Now let us study the general case where $E \in \mathbb{R}^{m \times n}$. As stated in Section 2, saddle points exist iff

$$b \in \mathcal{R}(E), \ c \in \mathcal{R}(E^\top). \tag{G.1}$$

Assume $b = Eb'$, $c = E^\top c'$. One can shift the origin of $x$ and $y$: $x \to x - b'$, $y \to y - c'$, such that the linear terms cancel out. Therefore, the min-max optimization problem becomes:

$$\min_{x \in \mathbb{R}^m} \max_{y \in \mathbb{R}^n} x^\top E y. \tag{G.2}$$

The set of saddle points is:

$$\{(x, y) | y \in \mathcal{N}(E), x \in \mathcal{N}(E^\top)\}. \tag{G.3}$$

For all the first-order algorithms we study in this paper, $x^{(t)} \in x^{(0)} + \mathcal{R}(E)$ and $y^{(t)} \in y^{(0)} + \mathcal{R}(E^\top)$. Since for any matrix $X \in \mathbb{R}^{p \times q}$, $\mathcal{R}(X) \oplus \mathcal{N}(X^\top) = \mathbb{R}^p$, if the algorithm converges to a saddle point, then this saddle point is uniquely defined by the initialization:

$$x^* = P_E^\perp x^{(0)}, \ y^* = P_{E^\top}^\perp y^{(0)}, \tag{G.4}$$

where

$$P_X^\perp := I - X^\dagger X, \tag{G.5}$$

is the orthogonal projection operator onto the null space of $X$, and $X^\dagger$ denotes the Moore–Penrose pseudoinverse. Therefore, the convergence to the saddle point is described by the distances of $x^{(t)}$ and $y^{(t)}$ to the null spaces $\mathcal{N}(E^\top)$ and $\mathcal{N}(E)$. We consider the following measure:

$$\Delta_t^2 = ||E^\dagger E y^{(t)}||^2 + ||EE^\dagger x^{(t)}||^2, \tag{G.6}$$

as the Euclidean distance of $z^{(t)} = (x^{(t)}, y^{(t)})$ to the space of saddle points $\mathcal{N}(E^\top) \times \mathcal{N}(E)$. Consider the singular value decomposition of $E$:

$$E = U \begin{bmatrix} \Sigma_r & 0 \\ 0 & 0 \end{bmatrix} V^\top, \tag{G.7}$$

with $\Sigma_r \in \mathbb{R}^{r \times r}$ diagonal and non-singular. Define:

$$v^{(t)} = V^\top y^{(t)}, \ u^{(t)} = U^\top x^{(t)}, \tag{G.8}$$

and equation G.6 becomes:

$$\Delta_t^2 = ||v_r^{(t)}||^2 + ||u_r^{(t)}||^2, \tag{G.9}$$

with $v_r$ denoting the sub-vector with the first $r$ elements of $v$. Hence, the convergence of the bilinear game with a singular matrix $E$ reduces to the convergence of the bilinear game with a non-singular matrix $\Sigma_r$, and all our previous analysis still holds.

