# OpenReview forum: "Convergence of Gradient Methods on Bilinear Zero-Sum Games"
_ICLR.cc/2020/Conference — Accept (Poster)_

### Official Review · AnonReviewer2 · 2019-10-23
**Official Blind Review #2**

**Rating:** 6

**Review:**

Summary:
The paper presents exact conditions for the convergence of several gradient based methods for solving bilinear games. In particular, the methods under study are Gradient Descent(GD), Extragradient (EG), Optimizatic Gradient descent (OGD) and Momentum methods. For these methods, the authors provide convergence rates (with optimal parameter setup) for both alternating (Gauss-Seidel) and simultaneous (Jacobi) updates.

Comments:
The paper is well written and the main contributions are clear.
I find the theoretical results of the paper interesting and promising, however i believe that the proposed analysis will be difficult to extend beyond bilinear games to more practical scenarios as the authors claim.  In particular, the proposed analysis is based on understanding the bilinear game dynamics using spectral analysis. This approach is not novel. It is well known for bilinear games (see for example [1] and the references therein) and is not easy to extent to general games.

The authors provide necessary and sufficient conditions under witch all previously mentioned algorithms (GD, EG, OGD,Momentum) converge for bilinear games. The convergence analysis  (Theorems 3.1 -3.4) is easy to follow and seems correct.

Main issue: The authors mentioned in their abstract that "... and understanding the dynamics of (stochastic ) gradient algorithms for solving ...". In addition in their figures 4 and 5 they compare stochastic methods. However there is no convergence analysis on stochastic variants of the proposed methods. Note that in [2], it was shown that stochastic variants can prevent the convergence of standard game optimization methods, while their deterministic version converges.
If the goal of the last experiment is to show that the proposed methods and analysis can be extended to more interesting general settings then the algorithms analyzed in the paper should be used in the numerical evaluation and not their stochastic variants. The authors should be more clear (from the abstract) on what algorithms they study. The paper clearly focuses on deterministic (full gradient) methods.

To conclude I liked the paper and the theoretical analysis seems correct however i am not convinced that the results could be of interest for the ICLR community. It focuses only on simple bilinear zero-sum games and on deterministic methods for solving them.

Minor Comments:
page 2, bellow eq 2.2 biliner---> bilinear
page 3 bellow eq. 2.6: Cesari---> Cesaro
In caption of figure 2: replace the x-axis and y-axis with horizontal axis and vertical axis respectively.

References:
[1] Gidel, Gauthier, Reyhane Askari Hemmat, Mohammad Pezeshki, Remi Lepriol, Gabriel Huang, Simon Lacoste-Julien, and Ioannis Mitliagkas. "Negative momentum for improved game dynamics." arXiv preprint arXiv:1807.04740 (2018).

[2] Chavdarova, Tatjana, Gauthier Gidel, François Fleuret, and Simon Lacoste-Julien. "Reducing noise in gan training with variance reduced extragradient." arXiv preprint arXiv:1904.08598 (2019).

========= after rebuttal =============
   I would like to thank the authors for the reply. After reading their response and the comments of the other reviewers I  decide to update my score to "weak accept".


**Experience Assessment:**

I have read many papers in this area.

**Review Assessment: Checking Correctness Of Derivations And Theory:**

I carefully checked the derivations and theory.

**Review Assessment: Checking Correctness Of Experiments:**

I did not assess the experiments.

**Review Assessment: Thoroughness In Paper Reading:**

I read the paper at least twice and used my best judgement in assessing the paper.

---

> ### Author Response · Authors · 2019-11-15
> **response to review #2**
>
> Thank you for acknowledging our theoretical contribution. We would like to clarify our contribution and novelty. We agree that spectral analysis, being a classic tool in numerical analysis, has appeared in a number of recent works on analyzing bilinear games and simple GANs. However, our main contribution differs substantially from existing work. Our novelty lies on the new tools for studying the spectrum, such as the Schur's theorem. The study of convergence for alternating (Gauss-Seidel) updates was not clear or systematic before our paper. We give a new and systematic approach for studying GS updates, using our theorem 2.3. Applying Schur's theorem we give necessary and sufficient conditions for convergence, which is also novel. We prove that Jacobi momentum never converges and we prove that convergence of Jacobi OGD always implies convergence of GS OGD, both of which were not (completely) known before and appear to be significantly interesting. It is perhaps not fair to claim that "our results are well known for bilinear games" since all prior works we are aware of appeared in ICLR, ICML, NeurIPS in the last 2~3 years, and to our best knowledge none of them achieved our contributions outlined above.
>
> "To conclude I liked the paper and the theoretical analysis seems correct however i am not convinced that the results could be of interest for the ICLR community. It focuses only on simple bilinear zero-sum games and on deterministic methods for solving them. "
>
> This work should be interesting to the ICLR community. In fact, many new algorithms such as OGD and negative momentum were first verified on bilinear zero-sum games. For example, in the ICLR 2018 paper "Training GAN with optimism" by Daskalakis et al (Section 3 and 4), the authors studied bilinear games as their only theoretical justification. Also, it is a simple yet useful simplification of Wasserstein GAN, since it captures the cycling behavior (Appendix E, Nagarajan and Kolter 2017).  This simplification is also studied in, e.g., ICLR 2019 by Gidel et al (Section 3.1 and 3.2), and (Daskalakis et al., 2018; Gidel et al. 2019a, b; Liang & Stokes, 2019). Our work provides new tools for analyzing gradient algorithms on bilinear games, which serves as a first step for theoretical understanding of the training of GANs. Moreover, studying bilinear min-max optimization could potentially help in other applications, such as adversarial training (Madry et al 2018).
>
> Although we only presented results on bilinear games, our experiments, including the WGAN examples mentioned, showed that the insights we obtain from our analysis could potentially apply to non-bilinear cases as well, such as the comparison between simultaneous (Jacobi) and alternating (Gauss-Seidel) updates, and the speed-up obtained by generalizing current algorithms.
>
> "If the goal of the last experiment is to show that the proposed methods and analysis can be extended to more interesting general settings then the algorithms analyzed in the paper should be used in the numerical evaluation and not their stochastic variants. The authors should be more clear (from the abstract) on what algorithms they study. The paper clearly focuses on deterministic (full gradient) methods. "
>
> We agree that we analyzed non-stochastic algorithms and we should have been clearer in the abstract (we have already modified it). In the numerical evaluation, we implemented our *deterministic* algorithms in the first paragraph and it agrees with our theory. For the experiments of WGAN, we took large batch sizes to reduce the noise, so that our theoretical results would apply qualitatively. The experimental results show that we might extend our analysis to stochastic algorithms if the noise is small. The goal of the experiments of GMMs is to show that our analysis might even be extended to stochastic algorithms on non-bilinear games.
>
> All minor comments have been taken into account in our revision.

---

### Official Review · AnonReviewer1 · 2019-10-24
**Official Blind Review #1**

**Rating:** 8

**Review:**

*Summary*
This paper studies the convergence of multiple methods (Gradient, extragradient, optimistic and momentum) on a bilinear minmax game. More precisely, this paper uses spectral condition to study the difference between simultaneous (Jacobi) and alternating (Gau\ss-Seidel) updates. The analysis is based on Schur theorem and give necessary and sufficient condition for convergence.

*Decision*
This paper tries to study a phenomenon that has known a recent surge of interest due to the numerous practical minmax application: the impact of alternating updates in minmax optimization. This problem is challenging because most of the theoretical analysis techniques have been developed to analyse simultaneous updates.
I think that this paper has treated well the related work and underlines well its contributions.
Even if the main contribution are theoretical the authors of this paper provide some experiments to confirm that the theory provides some meaningful insights.
However, I have the concern that the study is still limited to bilinear example and thus it is not clear how to apply it to non-bilinear objective (even locally because since the Jacobian has pure imaginary eigenvalues, even locally the Jacobian may have eigenvalues with negative real part).
Moreover this work do not provides significantly better (factor 8 improvement in the constants) convergence rate that the one provided on the literature (Tseng 1995, Mokthari et al 2019, Gidel et al. 2019) to solve bilinear games. The contributions is more about proving new (interesting) analysis tools and showing a better robustness of GS updates with respect to hyperparameter tunning.
To me, it is an accept.

*Questions*
- In Theorem 4.1 and 4.2 you use the word ‘optimal exponent’. In what sense these algorithms are optimal ? Usually the sense of optimal is when you achieve the lower bound of convergence. Is it the case here ?
- Right after theorem 2.3 you claim that GS updates simply leads to a shift of index for $L_i$ setting $L_{k+1}=0$ but it seems to me that then $\lambda L_1$ is missing in the sum. Is it just an unfortunate oversight? Does it affect the proofs of you main theorems ? These results only marginally (improve by a factor 8) improve upon Moktari et al. (2019).

*Remarks*
- To me the J and GS conditions in Theorem 3.2, 3.3 and 3.4 are more Lemmas than Theorems: They are condition that are hard to interpret and they have small interest if one cannot conclude on simple conditions (such as the ones you actually provide in the Theorem). My point is that providing such complicated conditions without condition is only half solve the problem of convergence.  In that sense, in the discussion of Theorem 3.4, you are a bit unfair with Gidel et al (2019) since unlike them you do not provide any convergence rate. On one side theorem 3.4 gives an interesting characterization for the convergence (that leads to the fact that at least one of the beat has to be negative) but if it does not lead to any rate it is a less interesting result.
- In Table 1 and 2, $\alpha,\beta_1$ and $\beta_2$ have not been introduced. It is thus hard to get the most of them.
- Theorem 3.1 has been stated in previous works (like for instance Gidel et al 2019). You should not claim it as a contribution.

=== After rebuttal ===
I've read the authors's response.
I agree with the concerns raised by Reviewer 2 regarding the experimental part and whether or not such theoretical study (far from the practical aspect) are of interest to the ICLR community.
I think that that paper would be more suited for a theoretical venue such as AISTATS (or ICML) but I also think that this work remains of interest to the ICLR community.
I am not convinced by the fact that this analysis could be generalized beyond bilinear game with a local analysis  because even an arbitrarily small perturbation can transform a bilinear example into a non-stable equilibrium (with no hope of local convergence), take for instance
$$ \min_x \max_y -\epsilon \|x\|^2 +  x^\top A y + \epsilon \|y\|^2$$


**Experience Assessment:**

I have published in this field for several years.

**Review Assessment: Checking Correctness Of Derivations And Theory:**

I assessed the sensibility of the derivations and theory.

**Review Assessment: Checking Correctness Of Experiments:**

I assessed the sensibility of the experiments.

**Review Assessment: Thoroughness In Paper Reading:**

I read the paper thoroughly.

---

> ### Author Response · Authors · 2019-11-15
> **Response to review #1**
>
> Thank you for your thorough reading of our paper and your positive review!
>
> *Questions*
> "- In Theorem 4.1 and 4.2 you use the word ‘optimal exponent’. In what sense these algorithms are optimal ? Usually the sense of optimal is when you achieve the lower bound of convergence. Is it the case here ? "
>
> By optimal we mean given a generalized algorithm and its hyperparameter choice, the optimal rate we can obtain on bilinear games with different hyperparameter configurations of this algorithm.
>
> "- Right after theorem 2.3 you claim that GS updates simply leads to a shift of index for setting but it seems to me that then is missing in the sum. Is it just an unfortunate oversight? Does it affect the proofs of you main theorems?"
>
> This is a typo and it does not affect our proofs. We have fixed it.
>
> "- Theorem 3.1 has been stated in previous works (like for instance Gidel et al 2019). You should not claim it as a contribution. "
>
> Thank you for pointing out this reference. We have acknowledged their priority in our revised draft.
>
> "However, I have the concern that the study is still limited to bilinear example and thus it is not clear how to apply it to non-bilinear objective (even locally because since the Jacobian has pure imaginary eigenvalues, even locally the Jacobian may have eigenvalues with negative real part)."
>
> We want to clarify that our analysis based on linear dynamical systems and Schur's theorem does not depend on whether the Jacobian has pure imaginary eigenvalues. These tools can apply to local analysis. By expanding a function to the second-order and repeating the same approach using LDS, one could extend the current results to non-bilinear objectives. See e.g., https://arxiv.org/pdf/1806.09235.pdf for local analysis.
>
> "Remark on conditions in Theorem 3.2, 3.3 and 3.4"
>
> We agree that our sufficient and necessary conditions are harder to interpret. The value of including them is as follows: (a) they offer a complete picture; (b) an explicit enumeration of them allows us to make important conclusions, such as Jacobi momentum never converges; (c) while it is possible to simplify the conditions by taking special choices of the step sizes (as in previous work), we believe these common special cases may not actually be the most suitable case for practitioners, and the sufficient and necessary condition may prove useful for later studies as it defines the region to search for the optimal convergence rate. We did not include any specific rate in Section 3 because this section only concerns about convergence vs divergence while in Section 4 we offer explicit rates. We have acknowledged in our comment the rate derived in Gidel et al (2019). Note that the rate for Jacobi momentum is a rate to diverge as opposed to converge.

---

### Official Review · AnonReviewer3 · 2019-10-27
**Official Blind Review #3**

**Rating:** 3

**Review:**

1) Summary
The manuscript presents a theoretical convergence analysis of gradient-based saddle point algorithms to solve min-max problems with a bilinear objective function. In particular, the analysis covers block updates and joint updates.

2) Quality
The paper -- being a theoretical analysis rather than a new algorithm -- seems mathematically rigorous but lacks motivation and also explanation.

3) Clarity
The notation is pretty clear and the results seem convincing but the mathematical formulation in eq. 2.1 and related assumptions (E being invertible, b and c being) need to be better justified in order to make the paper accessible to a broader audience.

4) Reproducibility
The data is mainly synthetic and the codes for the update schemes are available. This should render the results reproducible.

5) Evaluation
The evaluation is on synthetic settings and the results on the convergence seem convincing but a subtantial optimization problem remains for future work.

6) Questions/Issues
  A) Not sure about the implications, isn't the set of saddle points in eq (2.2) equivalent to x=y=0? If this is intended, there needs to be some explanation.
  B) It is not clear how well the results of the bilinear setting are applicable to the general case of bivariate functions. E.g. below eq (2.5) and below eq (5.1). There needs to be more justification.
  C) The start of Section 2 kicks off a little rough i.e. the formal setting could be better motivated.

7) Details
  a) Section 2, below eq 2.2: "biliner games" -> "bilinear games"
  b) Capitalization in references: Nash, GAN, Potenzreihen, Einheitskres
  c) References: "méthode iterative de résolution d'une équation variationelle"

**Experience Assessment:**

I do not know much about this area.

**Review Assessment: Checking Correctness Of Derivations And Theory:**

I did not assess the derivations or theory.

**Review Assessment: Checking Correctness Of Experiments:**

I assessed the sensibility of the experiments.

**Review Assessment: Thoroughness In Paper Reading:**

I read the paper at least twice and used my best judgement in assessing the paper.

---

> ### Author Response · Authors · 2019-11-14
> **Response to review #3**
>
> Thank you for reviewing our paper and providing critical comments. We would like to clarify a few points:
>
> "2) Quality
> The paper -- being a theoretical analysis rather than a new algorithm -- seems mathematically rigorous but lacks motivation and also explanation."
>
> We do have new algorithms as presented in Section 2. Although we retain the same names, these algorithms are more generalized than their original versions. In Section 4, we have shown that these new algorithms may behave better.
>
> Regarding the motivation, we want to stress that part of our motivation is based on the current trend of understanding the dynamics of training GANs. The bilinear game is an important example as it captures the cycling behavior of the learning dynamics when training GANs (Appendix E, Nagarajan and Kolter 2017; Section 2.2 of arXiv:1801.04406v4 (Mescheder et al. ICML 2018)). For this reason, it is a first step towards understanding the learning dynamics of GANs, as also analyzed in, for example, (Daskalakis et al., 2018; Gidel et al. 2019a, b; Liang & Stokes, 2019).
>
> This work should be interesting to the ICLR community. In fact, many new algorithms such as OGD and negative momentum are first verified on bilinear zero-sum games. For example, in the ICLR 2018 paper "Training GAN with optimism" by Daskalakis et al (Section 3 and 4), the authors studied bilinear games as their only theoretical justification. Also, it is a simple yet useful simplification of Wasserstein GAN, as studied in, e.g., ICLR 2019 by Gidel et al (Section 3.1 and 3.2). Our work provides new tools for analyzing gradient algorithms on bilinear games, which serves as a first step for the theoretical approach of understanding the training of GANs. Moreover, studying bilinear min-max optimization could potentially help in other applications, such as adversarial training (Madry et al 2018).
>
> Although we only study bilinear games theoretically, our experiments, including the WGAN examples mentioned, show that the insight we obtain from our analysis could potentially apply to non-bilinear cases as well, such as the comparison between simultaneous (Jacobi) and alternating (Gauss-Seidel) updates, and the speed-up obtained by generalizing current algorithms.
>
> "3) Clarity
> The notation is pretty clear and the results seem convincing but the mathematical formulation in eq. 2.1 and related assumptions (E being invertible, b and c being) need to be better justified in order to make the paper accessible to a broader audience. "
>
> Thanks a lot for your positive comments. Regarding the assumptions, we have justified the linear terms (b and c) in footnote 1 and added a discussion about general E (not necessarily invertible) in Appendix G.
>
> We have modified our draft to address your valuable comments 6A), 7a), 7b) and 7c). For 6C), we have added a few sentences about our motivation at the beginning of Section 2. For the comment 6B), the experiments in Section 5 show that our bilinear results might be extended for more general bi-variate functions. The experiments of WGAN show that we can extend our analysis to stochastic algorithms if the noise is small and the function is bilinear locally. The experiments of GMMs imply that our analysis might even be extended to stochastic algorithms on non-bilinear games.

---

### Author Response · Authors · 2019-11-14
**Draft updated**

To all reviewers: thank you for your comments and we have revised our draft accordingly. To help you track the changes easily, we colored the revised text in red.

---

### Decision · Program_Chairs · 2019-12-19

**Decision:**

Accept (Poster)

**Comment:**

All reviewers found the work interesting but worried about the extension to non-bilinear games. This is a point the authors should explicitly address in their work before publication.